# Reducing Energy Consumption in a Poultry Farm by Designing and Optimizing the Solar Heating/Photovoltaic System

Mansour Jalali [1], Ahmad Banakar [2,*], Behfar Farzaneh [1] and Mehdi Montazeri [2]

1 Department of Mechanical Engineering of Biosystems, Eghlid Branch, Islamic Azad University, Eghlid 73818-85939, Iran
2 Department of Mechanical Engineering of Biosystems, Faculty of Agriculture, Tarbiat Modares University, Tehran P.O. Box 14311-336, Iran
* Correspondence: ah_banakar@modares.ac.ir

**Abstract:** A solar heating system is designed to reduce energy consumption in a poultry farm. According to the physics and conditions of the indoor environment of the poultry building and the effect of the poultry weather conditions, the amount of $1.37 \times 10^8$ kJ/h during the year energy is required for heating. Then, by using double-glazed windows and insulation for the exterior walls of the building in the building architecture section, the amount of energy consumption is drastically reduced, and the required annual gas consumption is equal to 11,833 m$^3$. The surface required for the collector is recommended to supply 50% of the energy from the sun with the rest from the hybrid system. The results showed that 26 m$^2$ of a solar collector with an optimal slope of 45 degrees, and a tank volume of 440 L and a pump discharge of 1700 kg/h are required to provide 100% of energy. To receive the maximum amount of solar energy (maximum solar fraction (SF)), a collector surface equal to 30 m$^2$ is required. However, when the economic point of view is considered, the collector surface equivalent to 26 m$^2$ is recommended. To establish a balance, that is, 50% of the energy from the auxiliary system and the rest from the solar system, between the use of solar energy and the use of the auxiliary system, a collector area of 16 m$^2$ is needed. Based on this, 60 photovoltaic modules, which are 10 cells in series in 6 parallel circuits, is the most optimal mode.

**Keywords:** poultry farm; energy demand; TRNSYS simulation; solar collector; energy modeling

## 1. Introduction

The climate around a poultry farm is one of the variables that affect poultry farming performance. Parameters, such as temperature, humidity, wind, and sunlight, are especially important in their absolute value, and their daily and seasonal variations. The heat produced by the chicken's metabolism warms the air. This effect is useful in winter to keep the poultry warm. Latent heat, or moisture, is produced in two forms: the vapor of air from the breath and water in the urine [1]. The proper temperature for poultry in an incubation period strongly depends on the age of the poultry. The required temperature at the surface where the hens are located is 90 °F (about 32.2 °C) in the first three days of hatching. Due to the vital need of heating for day-old chicks, keeping the temperature in this range is critical for the chicks. After that, for each passing day, 1 °F (0.6 °C) should be reduced from the air temperature [2]. Finally, in the fourth week, it reaches about 70 °F (21.1 °C), then remains constant [3]. Excess heat is needed when the birds do not produce enough heat to maintain the right temperature in confined poultry farms in cold climates; which accounts for the largest share of the building's energy consumption. Thus, it is becoming increasingly clear that poultry farms need more insulation to maintain heat in cold climates [4]. Breeders need to know the energy consumption of their poultry house and carefully predicted the amount of energy consumption for ventilation, heating, and cooling, which for some breeding systems is a major indicator of energy consumption. This is because at every stage of a bird's development, there is an optimum temperature at which the birds produce the best in

terms of using feed energy to grow. If the birds are kept at a temperature below the desired temperature, the birds increase their feed absorption and use more feed energy to keep their bodies warm, which increases the amount of production and reduces meat production. On the other hand, if they are kept at a temperature above the desired temperature, they reduce the amount of food consumed to limit heat production, which in turn leads to less meat production. Therefore, an efficient heating system is necessary for every broiler house to maintain the required temperature throughout the year [5]. The use of renewable energy is a suitable solution to solve these problems [6]. The impact of alternative energy programs on the feasibility of solar PV systems in several solar regions in the poultry industry was examined. The results were financially beneficial [7]. In an experimental study, poultry performance was evaluated in terms of relative humidity, ammonia concentration, poultry production, feed conversion rate, required power, and production cost. Experimental results reveal that the optimum conditions for enhancing the poultry production (2.29 kg) with conversion rate (1.45 kg feed per kg gain), ammonia concentration at the fifth week (13.65 ppm), and production cost (1.12 US \$/kg) are achieved by using a power operating system of flat-plate solar collector integrated with photovoltaic under 2 min fan stopping periods [8].

Masry et al. [9] investigated the effect of a heating system powered by renewable energy on poultry houses. The results showed that using Biogas increases poultry body weight/bird after 6 weeks by about 3%, and by about 4% approximately in total body weights, and increases production efficiency factor (PEF) by about 10.3% compared to a conventional system.

Zeyad et al. [10] investigated the economic feasibility analysis of a designed poultry farming zone with renewable energy resources in Bangladesh. The results showed that from the load profile, the peak load was found 26.12 kW, and the average daily consumption was found 226.36 kWh. Furthermore, renewable energy resources, such as solar and biomass energy, were utilized to develop a smart microgrid to fulfill the energy demand of the poultry farm.

Cuie et al. [11] investigated the energy, economic, and environmental assessments on hybrid renewable energy technology applied in poultry farming. The results showed that the electrical energy production from the photovoltaic array could reach 11,867 kWh per annum, whereas the heat pump thermal output is about 30,210 kWh per annum. Meanwhile, the overall gas and electrical cost of the hybrid renewable heating system are £320 and £129, respectively, which is much less than that of the gas burners system and could save £763 and £750, respectively, resulting in less than a 6-year of payback period.

In recent years, simulation methods have received much attention due to saving time and money [12]. A transient model has been designed using the TRNSYS 16.1 software to evaluate the performance of different greenhouses, and the accuracy of the proposed model is confirmed using measured data in the reference greenhouse (conventional greenhouse) [13]. Computer simulation makes it possible to predict the hourly temperature and relative humidity inside the breeding unit for each year [14]. Dynamic simulation models are needed to predict the energy consumption of climate control systems used in livestock houses [15]. Dynamic simulation models make us consider sudden changes in boundary conditions, which are usually preferred by self-made computational models that use ready-made simulation tools (e.g., TRNSYS and Energy Plus 8.4.0) due to the availability of ready-to-use tools for accurate determination of thermal behavior in livestock and boundary conditions. In addition, the degree of customizability of self-made models is high and, therefore, it has been proven that they are more compatible with specific goals [16].

Lee et al. [17] investigated the adequate capacity of ground source heat pumps in energy-saving pig farms using building energy simulation. The results showed that the ground source heat pump's total cost of ownership was the cheapest, but the installation cost was the highest.

Gargab et al. [18] investigated the energy efficiency for social buildings in Morocco by TRNsys simulation. The results showed that the glazing solution does not exceed 6% in

savings on total energy demand, including the electrical demand for domestic hot water, while insulation reaches 17% in savings in zone 4.

Lee et al. [19] investigated the dynamic energy modeling for analysis of the thermal and hygroscopic environment in a mechanically ventilated duck house. The results showed errors of 1.71% and 4.33% for the air temperature and relative humidity, respectively.

One of the most important parameters to be considered in a poultry house is the determination of heat loss and heat increase. Heat loss generally involves losses from walls, floors, and ventilation [20]. Considering the economic approach, the strong role of energy is also tangible in other industries, and, due to its importance, it has challenged researchers [21–23]. According to the review of the previous articles, the simulation of the use of underfloor heating systems and its optimization has not been completed, as a result, in this research, a solar heating system is designed to reduce energy consumption in a poultry farm (with a capacity of 300 pieces) and the thermal load of the building is transiently simulated in TRNSYS software. By using this system, the amount of energy consumption is greatly reduced. Additionally, the heat produced by the body of chickens, with the change of weight and its effect on the heat load of the building with the growth process of the chicken, is investigated. The solar water heater in this system is optimized for heating the hall using the floor heating system. In this study, different solar thermal collectors are investigated and the best solar collector surface is suggested to achieve maximum efficiency of solar energy.

## 2. Materials and Methods

### 2.1. Physics of the Problem

This research is a thermal simulation of a broiler farm in Ardestan city. Ardestan is located at 33 degrees and 23 min north latitude and 52 degrees and 22 min east longitude relative to noon Greenwich. Figure 1 shows the location of Ardestan city [24].

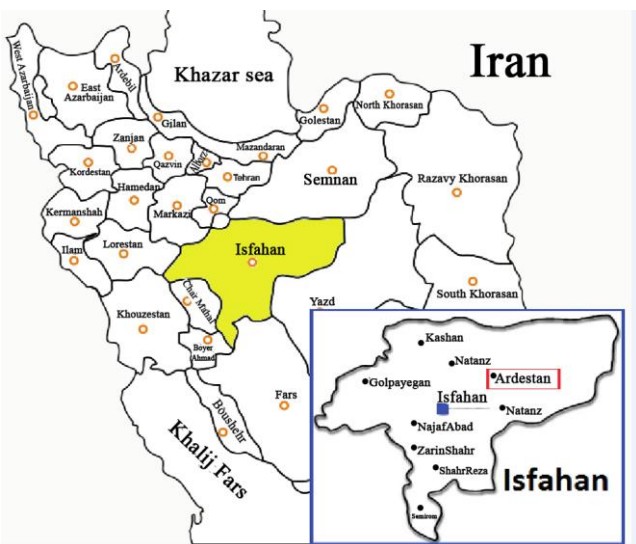

**Figure 1.** The location of Ardestan city.

Figure 2 shows the physics of the present problem. The dimensions of the poultry hen house are 5 m wide and 8 m long. There are 2 inlet air vents (windows) with the dimensions of 1.90 by 1.6 m. The average height of the roof is 2.5 m and, in poultry houses with a sloping roof, the roof is covered with plastic cartons, glass wool, and metal sheets. The location of the poultry farm is east–west (due to the local wind) and the fans (hen house aerators) are in the west direction. The window is made of ordinary single-walled metal and the door is made of metal. The materials used in conventional poultry farming are 20 cm walls and use of black cement on both sides and the poultry floor is also cement. The existing heating system is the jet heater type, the number of which is in the hen house.

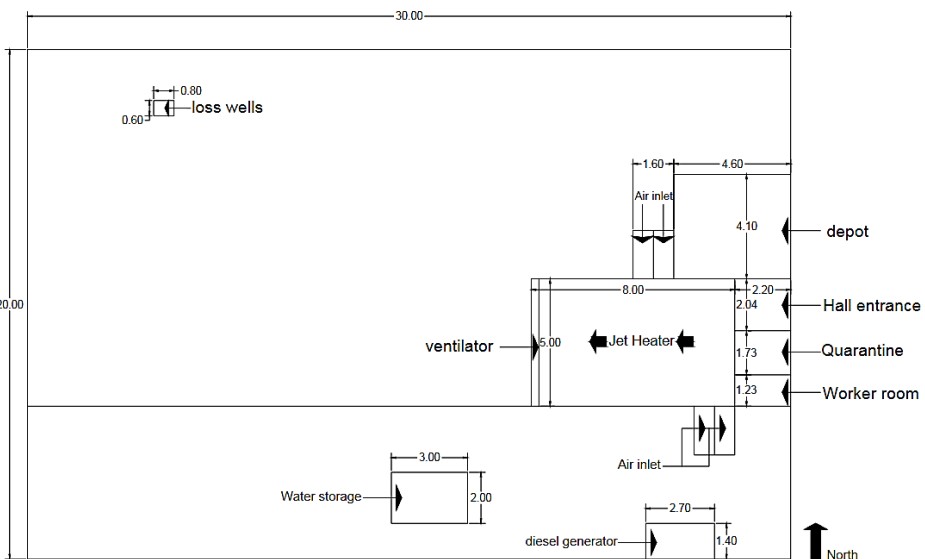

**Figure 2.** The schematic of the present problem.

*2.2. Mathematical Description and Solution Method*

The mathematical model of indoor heat loads is based on energy and mass balance. For transient conditions, the equation can be written as follows [25]:

$$mC_P \frac{dT_i}{dt} = q_{sup} + Q_s - Q_v - Q_w - Q_f \tag{1}$$

Birds, like all warm-blooded animals, produce heat, moisture, and carbon dioxide, which are by-products of their vital activities. In all stages, the body temperature is kept approximately in the range of 40 °C to 42.8 °C. As the temperature of the hall is usually lower than this temperature, the birds continuously lose heat. If the energy to compensate for the lost heat is not available to the bird, the body temperature will drop, and the bird cannot survive for a long time. Although laying hens may lay eggs above the temperature range of 12.8 °C to 23.9 °C at maximum production, the appropriate temperature is probably closer to 23.9 °C [26]. Researchers have reported that for every 5 °C increase in ambient temperature, egg laying increases from 15 °C to 30 °C. The relative humidity at the highest temperature was 50% or less. The feed conversion factor (kilogram of feed per dozen eggs) has increased by 20% at the highest temperature because less energy from food is needed to maintain body temperature. Egg size and eggshell thickness decreases at high temperatures, and the producer must maintain the ambient temperature in a way that is economically feasible. For example, in winter, reducing feed conversion efficiency at colder ambient temperatures is less expensive than providing additional heat in laying hen halls [27].

Heat production for chickens in food deprivation and rest is approximately 2.75 calories per hour per gram of live weight. For each 1.82 kg chicken, this amount is equal to 5000 calories or 20 BTU per hour. Normal activity increases heat production and increases food consumption [28].

Due to the variability of activity and the amount of daily food consumption, two-way changes in basic heat production, which is maximum at 8 in the morning and minimum at 8 in the afternoon, and the decrease in heat production due to the decrease in activity during the dark period, the calculation of heat production for each chicken is approximate. is. In addition, when the air in the poultry house is warm, there is no heat loss due to the latent heat of evaporation in the breathing air. Therefore, heat production decreases. Of course, in winter, if the temperature of the hall reaches 1.1 °C to 4.4 °C, approximately 20% of the total heat production, it is due to this reason, and when the temperature of the hall reaches 2.8 °C, each chicken emits 30 BTU per hour [28].

One of the most important parameters that must be taken into account in the poultry breeding hall is the determination of heat loss and heat gain. Heat loss generally includes losses from walls, floors, and ventilation. Sources of heat gain include heat gain from chickens, solar heat absorption, and lighting [29].

First, the optimal thermal conditions inside the poultry house are described according to the age of the birds. Then, the relationships for calculating the thermal benefits received by the poultry house, such as the thermal benefits caused by the birds, are stated. Then, the equations and relationships needed to calculate the heat losses of the building are presented, and, finally, the heat load of the hall is calculated according to the characteristics of the building of the poultry hall, the environmental conditions, and the optimal thermal conditions inside. Figure 3 schematically shows the thermal energy balance for a poultry house [30].

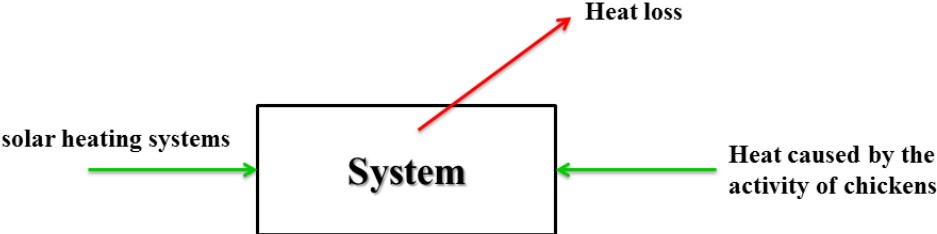

**Figure 3.** Schematic of thermal energy balance in the poultry house.

Through Equation (2), the total heat produced by poultry can be obtained, which includes tangible heat and latent heat. The total heat generated by the chicken can be determined as follows [31]:

$$Q_{chicken} = 10.m_b^{0.75} \left[4 \times 10^{-5} \times (20 - T_i)^3 + 1\right] \tag{2}$$

where $Q_{chicken}$ is heat produced by the chicken, and $m_b$ and $T_i$ are chicken mass [kg] and indoor temperature [K], respectively.

To calculate tangible heat ($Q_{sens}$), which is part of the total calculated heat of Equation (2), we can use Equation (3):

$$Q_{sens} = Q_{chicken} \left(0.8 - 1.85 \times 10^{-7}(10 + T_i)^4\right) \tag{3}$$

In poultry houses, the main losses are from walls, doors, windows, ceilings, and floors, and due to the heat load due to air conditioning. Conductive heat transfer through building walls, such as walls, ceilings, doors, windows, and glass, can be calculated from Equation (4):

$$Qc = \frac{\sum_c (UA)_C \cdot (T_i - T_o)}{A_f} \tag{4}$$

where U is the overall heat transfer coefficient [W/mK], $A_f$ is wall and door surface [m$^2$], and $T_o$ is the outdoor temperature [K].

The overall heat transfer coefficient is determined as follows:

$$U = \frac{1}{R_t} \tag{5}$$

where

$$Rt = Ris + R1 + R2 + R3 + \ldots + Ros \tag{6}$$

where $R_{is}$, and $R_{os}$ were the thermal resistance unit of the inner layer surface area [(m$^2$·K)/W] and the thermal resistance unit of the outer layer surface area [(m$^2$·K)/W].

The thermal resistance of the unit surface of each layer is determined as follows [32]:

$$R_{1,2} = \frac{X_{1,2}}{K_{1,2}} \tag{7}$$

where $K_{1,2}$ is the thermal conductivity of each layer $[W/mK]$ and $X_{1,2}$ is the thickness of each layer of the building $[m]$.

The rate of thermal conductivity transfer from the floor in winter is determined as follows [33]:

$$Qf = F.P(Ti - To) \tag{8}$$

where F and P are experimentally constant factors of ambient heat loss $[W/mK]$ and building environment $[m]$.

The conductive heat transfer through building walls, such as walls, ceilings, doors, windows, and glass, can be calculated from Equation (9) [22]:

$$P = C_i \rho_i q_v (T_{in} - T_{out}), \; P = \text{ventilation heat loss} \tag{9}$$

where $C_i$, $\rho_i$, and $q_v$ are specific heat of the air $[J/kgK]$, density of air $[kg/m^3]$, total volumetric rate of airflow $[m^2/s]$, respectively.

The calculations of the total heat loss are equal to the sum of the losses expressed and are obtained as follows:

$$Q_{tot} = Q_W + Q_V + Q_f = BLC(T_i - T_o) \tag{10}$$

$Q_w$, $Q_v$, and $Q_f$ are heat loss from walls and ceiling $[W]$, heat loss by the ventilation system $[J/kgK]$, and heat loss from the floor of the building $[W]$, respectively.

Where BLC represents the Building Load Coefficient of the building and is calculated from Equation (11):

$$BLC = \sum_c UA + \sum F.P + \sum \dot{m}_a.c_p \tag{11}$$

According to the calculation of heat loss from Equation (10) and the calculation of thermal gain from Equation (3), the required heat load can be calculated from Equation (12):

$$Q = Q_{tot} - Q_{sens} \tag{12}$$

It should be noted that some sources have suggested that the total heat load required to be multiplied by a factor of 1.4 to 1.6 to account for existing uncertainties and errors [33].

The Internal Rate of Return (IRR) is the discount rate that zeroes the Net Present Value (NPV) for all cash flows of a project using the NPV formula [26]:

$$NPV = \sum_{t=1}^{t} \frac{C_t}{(1+R)^t} - C_0 \tag{13}$$

To calculate the IRR, it is necessary to set the NPV to zero and obtain its discount rate. In other words, we can write [26]:

$$0 = \sum_{t=1}^{t} \frac{C_t}{(1+IRR)^t} - C_0 \tag{14}$$

The SF is calculated as follows [27]:

$$\text{solar fraction} = \frac{E_s}{E_T} \tag{15}$$

That $E_T$ is the total energy required is the sum of the total solar energy received and is auxiliary energy.

The steps are summarized in the form of a Flowchart (Figure 4), which shows the steps of conducting studies in this section.

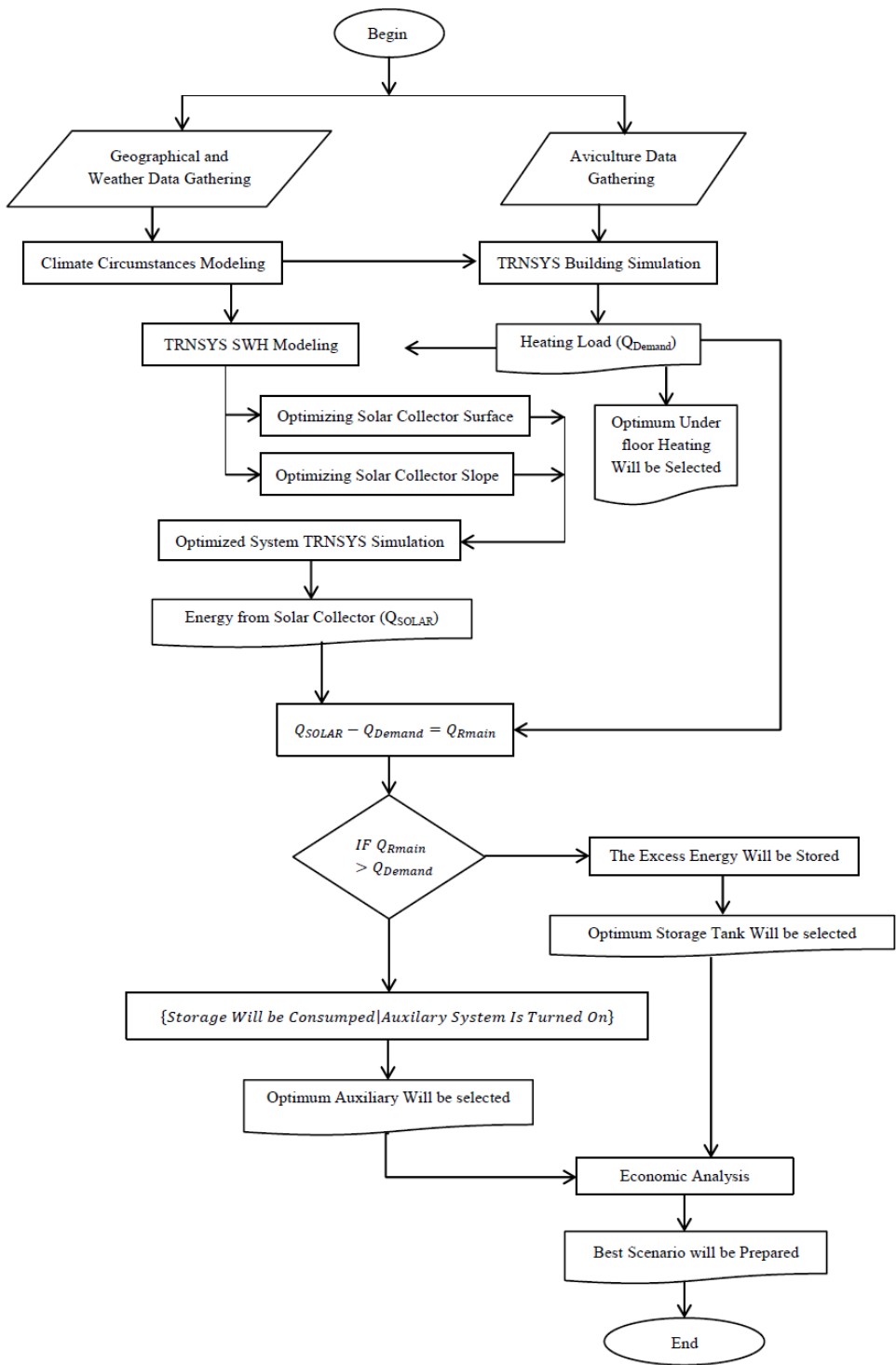

**Figure 4.** Flowchart of the research process in thermal modeling.

The simulation and optimization process of the solar electric system is shown in Figure 5.

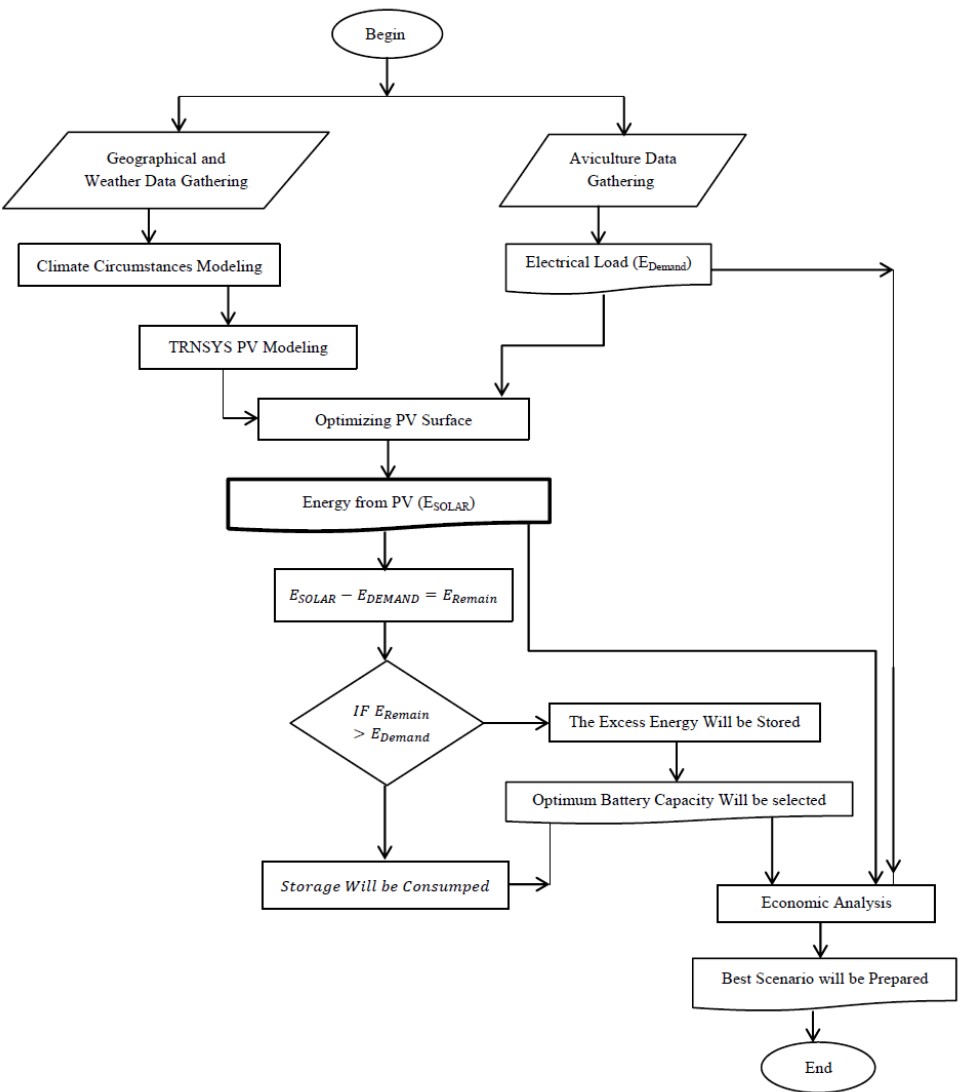

**Figure 5.** Flowchart of the simulation and optimization process of the solar electric system.

The basis of model used in this simulation is based on the equivalent circuit model of five parameters proposed by Duffy and Beckman [32]. This model can be solved by specifying the I-V diagram by the manufacturer of short-circuit current and open-circuit voltages in this study; it is assumed that poultry farming starts in the first hour of January. Thus, a poultry cycle takes place once every 56 days. Given that the time between each cycle is 15 to 20 days, in this study, we consider the interval between 2 cycles to be 17 days. Thus, the operation profile of the building in Figure 6 is presented hour by hour during the year. In Figure 6, the number one indicates activity and the number zero indicates the time between two cycles. Considering the complete information about the distribution of the desired temperature in the poultry house about the age of the chickens and the activity of the poultry farm, the following profile is presented hour by hour during the year about the age of the chickens and the activity of the house. As shown in this curve, the optimum temperature for keeping chickens in the first 3 days is 32.2 °C. This temperature decreases gradually during the days of operation until it reaches a constant temperature of 21.1 °C on the 21st day. The results are following Dbouk et al. [34]. In the time interval between the two operations, since the indoor temperature is not controlled and the heating system is turned off, and due to the disinfection of the indoor environment, the indoor doors are opened, and the indoor temperature is assumed to be equal to the outside temperature.

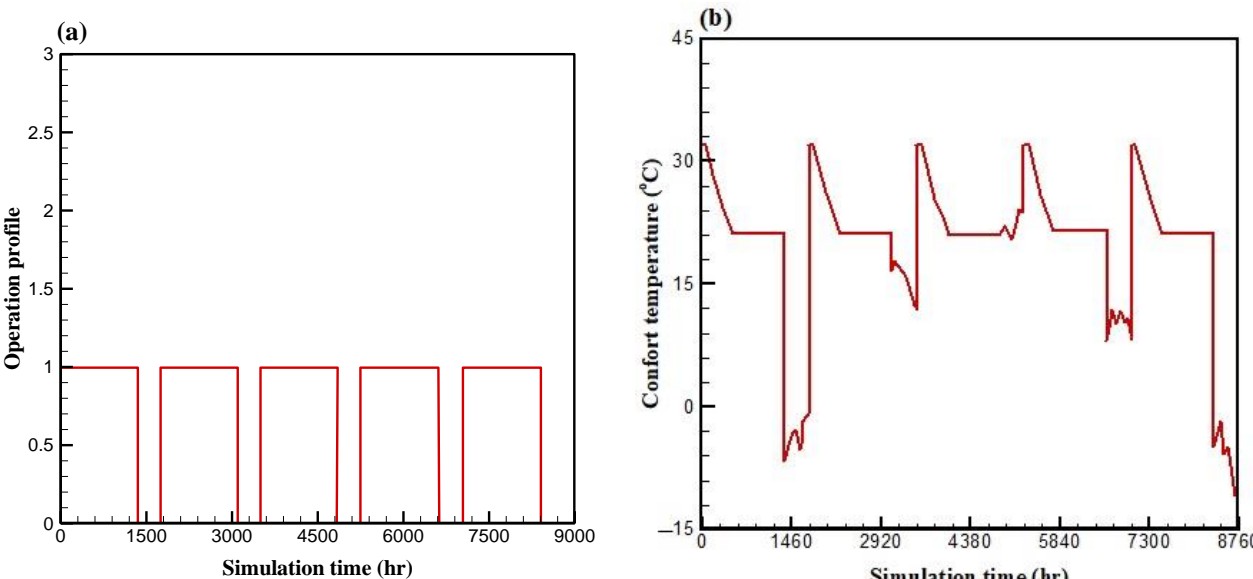

**Figure 6.** Operation profile of the building (**a**) Optimal temperature distribution in the poultry house about the age of the chickens (**b**).

### 2.3. Calculation of Heat Gain and Moisture Production of Chickens throughout the Year

Another important parameter in this simulation is the calculations related to the heat production of chickens throughout the year in relation to their age and the time of operation of the poultry house. The thermal gain of the chicken in each cycle is calculated and presented in Figure 7.

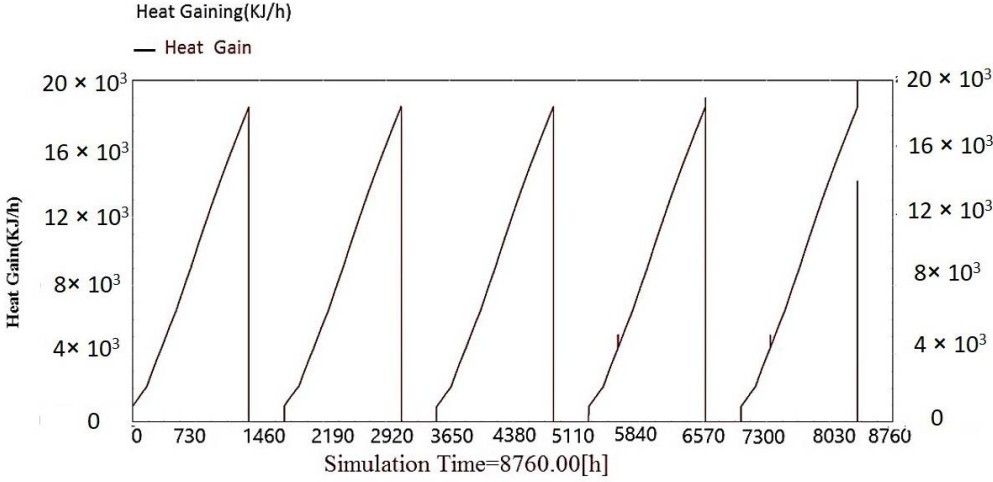

**Figure 7.** Poultry production temperature profile in the hall throughout the year, hour by hour, with increasing growth age.

This information was also evaluated for the production of moisture caused by the breathing of chickens, and the result is presented in Figure 8.

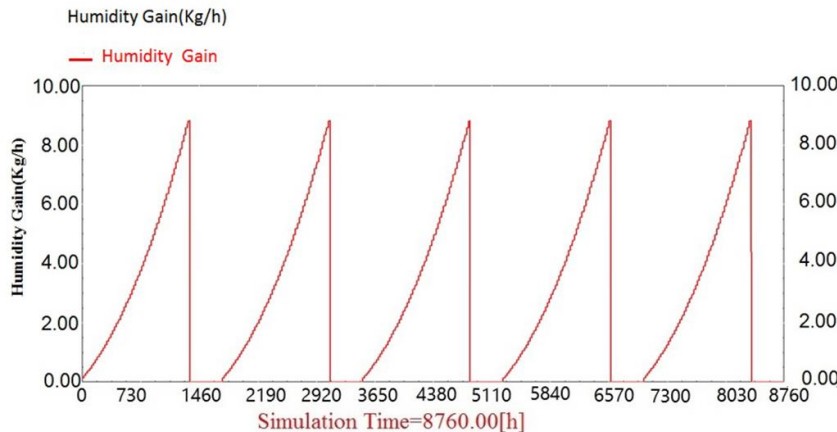

**Figure 8.** Poultry production humidity profile in the hall throughout the year hour by hour with increasing growth age.

For thermal calculations of the system in the TRNSYS software, according to the mentioned cases, the values of heat losses and heat load required for poultry are calculated on an hourly basis. These values are for the cold seasons of the year that require heating. Thus, by using type 88 of TRNSYS software (or similar cases), and defining the physics and coefficients of heat transfer of the building and the internal heat efficiency of the building, such as the heat produced by chickens, electrical equipment, etc., as an input to the model and values. The heat load and temperature inside the building are simulated hour by hour, and when the building temperature is lower than the standard temperature of broiler poultry, the amount of heat load is evaluated. As the analytical diagram provided by TRNSYS software is in terms of power at different hours, by integrating below the surface of the curve, we can estimate the energy required to heat the building. This integration is completed on a daily, monthly, and hourly basis for further analysis. By estimating the required power and energy, the underfloor heating system can also be evaluated and designed. First, there is a need to model the weather conditions of Ardestan city, hour by hour, in the form of a tmy2 file, which can be checked in TRNSYS software. For this purpose, the geographical location and the information collected from the Meteorological Organization to model the mentioned conditions is performed using Meteonorm software. Based on this, the longitude and altitude and the nearest synoptic stations (available in the software database) are modeled during the models that are present in the software by default, and the climatic information is provided hour by hour. To validate this information, the output of this software can be compared with the monthly average of the information provided by the synoptic station in the area.

Figure 9 shows the modeling of a poultry building in the current conditions. As shown in Figure 9, the climatic conditions of Ardestan city are defined as the entrance to the poultry building. To simulate the weather conditions, TRNSYS 16.1 software is used, which is a COMPONENT with the ability to read simulated weather conditions with the extension TMY2. One of the advantages of this component is the ability to calculate the intensity of scattered radiation with the PEREZ model. This model is one of the most accurate models to simulate the intensity of scattered radiation [35]. Since one of the required parts in the type 88 input of TRNSYS software is relative humidity, and the weather model of this software, which is capable of reading weather conditions, lacks relative humidity, by using a psychometric COMPONENT, Using the dry air temperature and the amount of humidity, the relative humidity has been evaluated on an hourly basis. The amount of heat generated by the chickens during each cycle, depending on their weight, is defined as the input using the DataReader COMPONENT. Other parameters affecting this section, such as humidity and heat generated by electrical equipment and lighting, are also defined in the form of a txt file to DataReader. All these conditions, including the ones analyzed in the previous section, have been entered as parameters and type 88 inputs that can calculate the thermal

load by the Lumped Capacitance method, and the conditions of the poultry building have been evaluated [36].

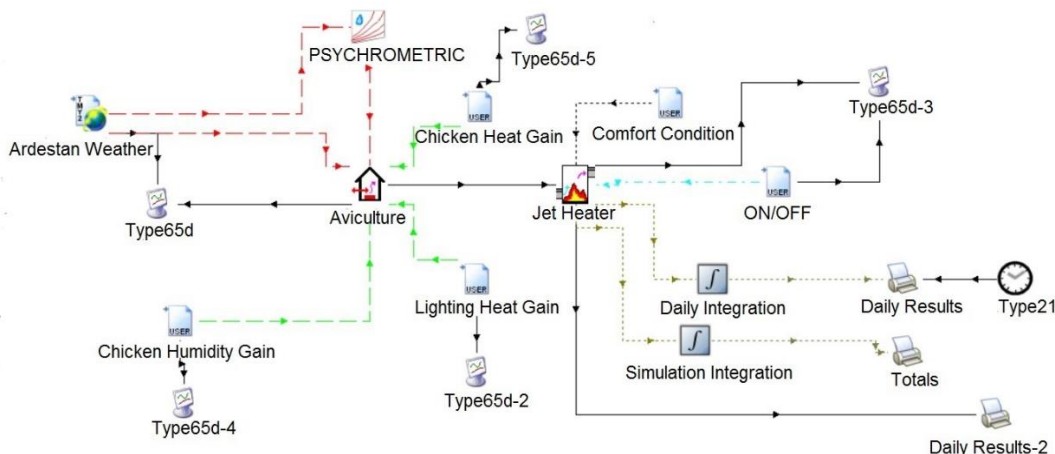

**Figure 9.** Simulated model in TRNSYS software for existing conditions.

## 3. Results and Discussion

Due to the clarity of the assumptions presented in the previous section, we can perform calculations related to the heat transfer coefficient. The climatic characteristics of the place are shown in Table 1. Table 2 shows the overall result of these calculations. Additionally, in Table 3, the thermal capacity calculations of the building are shown, and, to calculate the thermal capacity of the building, we first must calculate the surface mass of the roof, wall, and floor of the building. The surface mass of the building average mass is one square meter of the surface of the outer shell of the building. The information displayed in Table 3 with variables will be examined transiently and hour by hour. This information includes information and environmental conditions and is derived from weather conditions.

**Table 1.** Characteristics of climate and solar radiation in the place based on the data of Meteonorm software.

| Wind Direction Angle | Wind Velocity (m.s$^{-1}$) | Relative Humidity (%) | Temperature | |
|---|---|---|---|---|
| 7.62 | 0 | 9.89 | −8.88 | Minimum |
| 358.15 | 7.9 | 98.45 | 44.33 | Maximum |
| Solar angle | Solar elevation angle | The angle of the solar horizon | The intensity of sunlight on the horizon | |
| 10.03 | 59.72 | 10.06 | 378.15 | Minimum |
| 98.98 | 116.047 | 89.82 | 639.3 | Maximum |

**Table 2.** Total heat transfer coefficient.

| Total Coefficient | U (kJ/hr/m$^2$/K) | A (m$^2$) | UA (kJ/hr/K) |
|---|---|---|---|
| Floor Of The Building | 5.57 | 400.00 | 222.68 |
| Roof Of The Building | 6.73 | 400.00 | 269.15 |
| Exterior Walls | 8.57 | 54.92 | 470.74 |
| Window | 20.88 | 6.08 | 126.95 |
| Door | 20.88 | 4.00 | 83.52 |
| Total Coefficient | 8.09 | 145.00 | 1173.04 |

**Table 3.** Heat capacity of the building.

|  | A (m²) | Thickness (m) | Density (kg/m³) | Capacity (kJ/kgK) | Capacitance (kJ/kg) |
|---|---|---|---|---|---|
| Floor | 40 | 0.11 | 2200 | 1.2 | 11,616 |
| Roof | 40 | 0.0225 | 1800 | 0.8 | 1296 |
| Exterior Walls | 54.92 | 0.26 | 2080 | 1 | 29,700.736 |
| Air | 32 | 4.25 | 1.2 | 1 | 163.2 |
| Heat capacity of the building | | | | 42,775.936 | |

Figure 10a shows the indoor temperature without considering the jet heater (output type 88) and calculated by TRNSYS. As long as the indoor temperature reaches the desired temperature, the jet heater starts, then shuts off. The maximum indoor temperature is 37 °C, which occurs in the summer. The minimum temperature is −7 °C, which occurs in January.

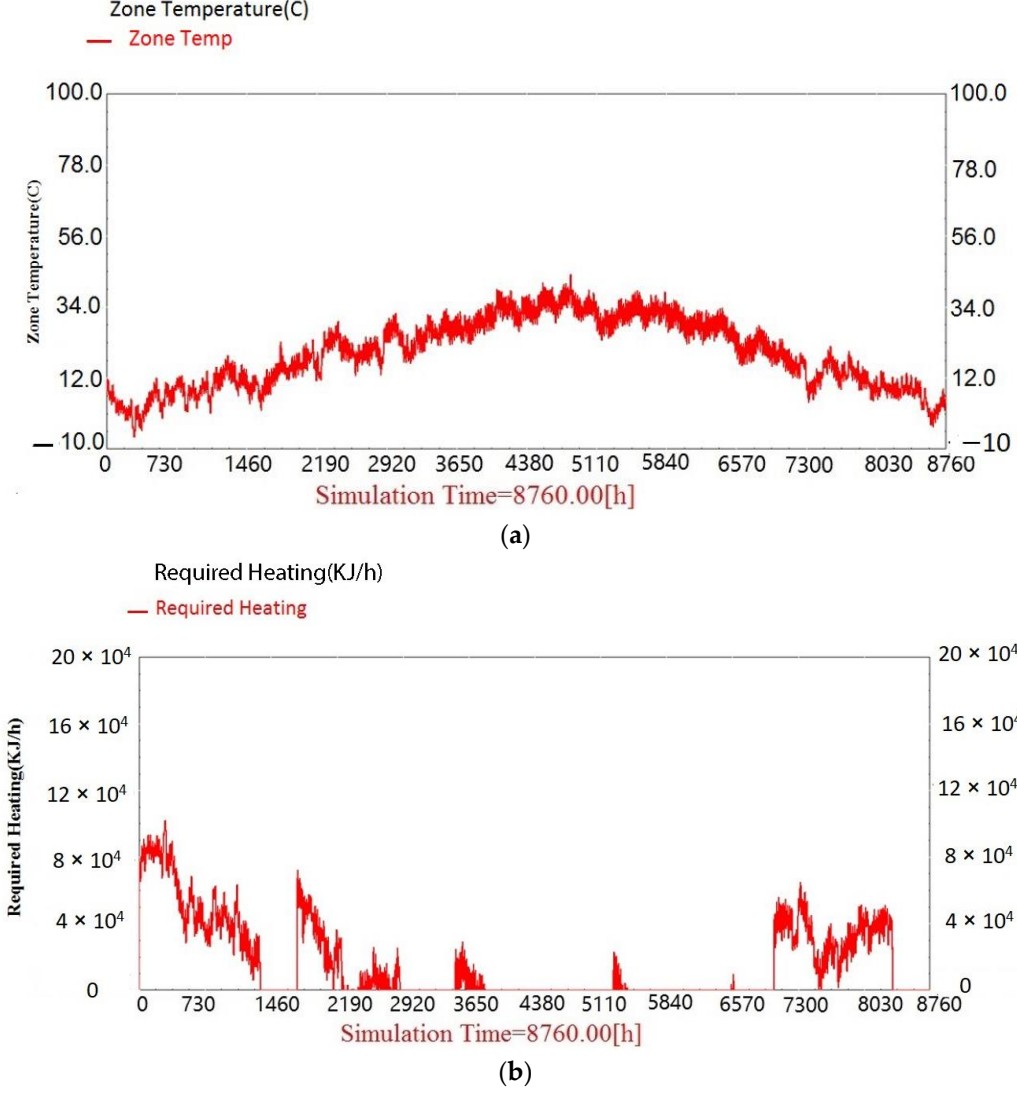

**Figure 10.** Indoor temperature in existing conditions (**a**) Indoor temperature required, simulated in TRNSYS software in existing conditions (**b**).

Figure 10b shows the heat required in the poultry house. As shown in Figure 10b, on 1 January, which is equivalent to the remainder of January, the building needs heat. This

heat requirement is met by a jet heater. Heat is needed as long as the chickens are indoors. When the chicken leaves the poultry house, the required amount of heat will be zero.

According to Figure 10 and integrating from below the power and time curve, the amount of energy required for the hen house for annual heating in the current conditions is equal to 137,462,334 kJ per year. Considering the value of thermal gas equal to 8000 kJ/m³, the amount of 17,182 m³ of gas per year is required.

$$\frac{137462334 \ (^{KJ}/_{hr})}{8000 \ (^{KJ}/_{m^3})} = 17182 \ m^3 \tag{16}$$

To validate and increase the validity of the amount of gas consumed during one year of operation in the poultry farm, it was referred to the amount of gas consumed in this poultry farm is reported to be 15,900 m³. The difference between these two numbers is equal to 1282 m³/year. This difference is approximately equal to 8%, and one of the reasons for the deviation is the lack of accurate adjustment of the comfort temperature of the chickens.

In addition, because the value of this deviation is approximately equal to 8%, the results are documented with acceptable accuracy and can be programmed.

To reduce energy losses from external walls, the use of thermal insulation and double-glazed windows has been proposed. Accordingly, the calculations are resumed. In this case, the amount of Building Loss Coefficient changes from 8.09 kJ/h to 5.1 kJ/h per square meter. The results of this evaluation show that if the heating system is not used for poultry, the temperature inside the poultry building in Ardestan city will change, as shown in Figure 11a. By investigating below the surface of Figure 11b, the amount of energy required for the annual heating of the hall is equal to 94,671,627.9 kJ per year, which is equal to 11,833 m³. Therefore, by performing these few simple operations, the annual savings will be 42,790,706 kJ per year (31%).

In the following, the economic analysis of these changes will be discussed. The cost of required insulation and the amount of insulation, and double glazing of windows and the total investment cost required are presented in Table 4.

**Table 4.** Cost savings for architectural solutions.

| Energy Consumption (kJ/year) | Optimal Energy Consumption (kJ/year) | Amount of Energy Saved (kJ/year) | Amount of Energy Saved (m³/year) | Global Gas Price ($/m³) | Amount of Energy Saved ($/year) |
|---|---|---|---|---|---|
| 137,462,334 ± 1% | 94,671,628 ± 1% | 42,790,706 ± 1% | 5349 | 0.101 | 52,960.396 ± 1% |

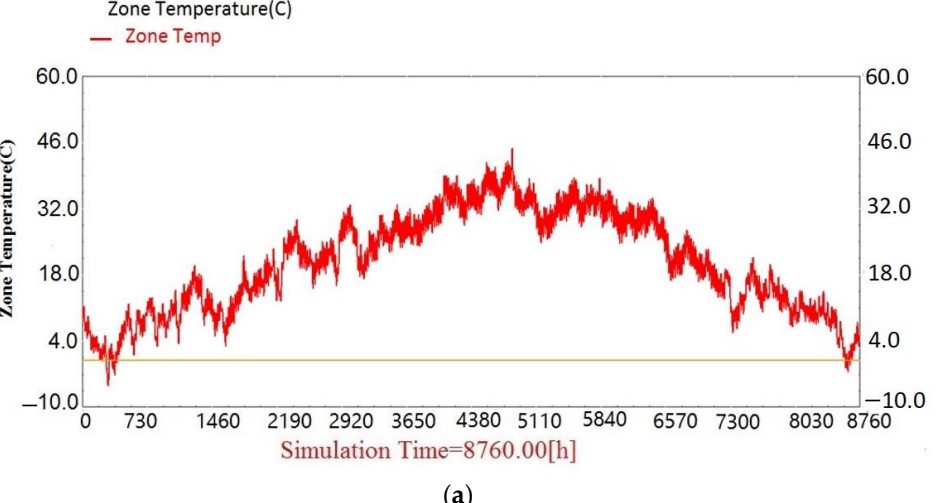

(a)

**Figure 11.** *Cont*.

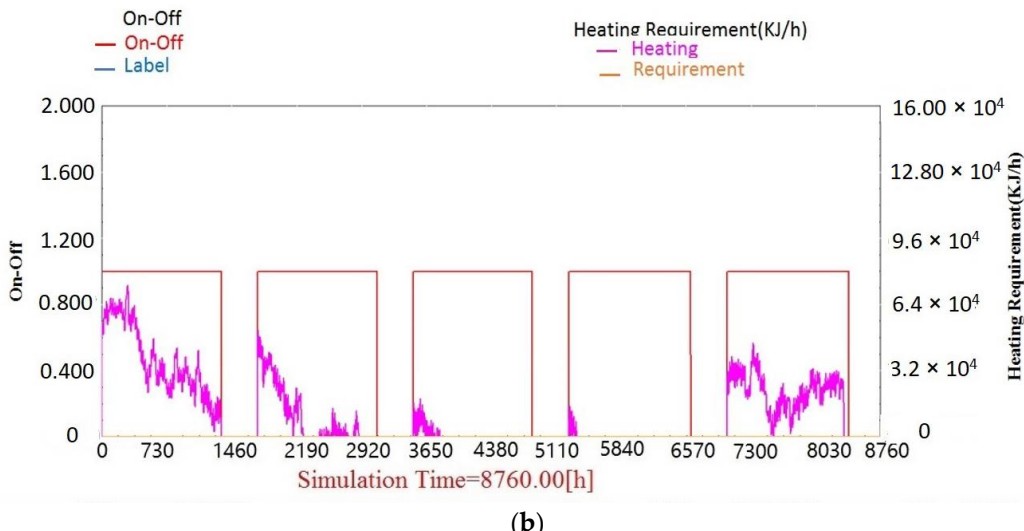

(**b**)

**Figure 11.** Indoor temperature in optimized conditions (**a**) Indoor temperature required, simulated in soft TRNSYS software in optimized conditions (**b**).

Therefore, the simple investment return of these solutions is equal to 3.6 years, which is considered a cost-effective solution from an economic point of view.

The cross-sectional selection of the solar collector is discussed. The assumptions for all the leading designs are presented in Table 5.

**Table 5.** Assumptions for solving the solar heating system problem.

| Description | Quantity | Unit |
| :---: | :---: | :---: |
| Component Type | Type 1b | - |
| Collector Type | Flat Plate Collector | - |
| Collector area | Variable | $m^2$ |
| Fluid specific heat | 4.19 | $kJ/kg·K$ |
| Tested flow rate | 40 | $kg/h·m^2$ |
| Intercept efficiency | 0.8 | - |
| Efficiency slope | 13 | $kJ/h·m^2·K$ |
| Efficiency curvature | 0.05 | $kJ/h·m^2·K^2$ |
| Optical mode 2 | 2 | - |
| First-order IAM | 0.2 | - |
| Second-order IAM | 0 | - |
| Inlet temperature | Variable | C |
| Inlet flowrate | Variable | kg/h |
| Ambient temperature | Variable | C |
| Incident radiation | Variable | $kJ/h·m^2$ |
| Total horizontal radiation | Variable | $kJ/h·m^2$ |
| Horizontal diffuse radiation | Variable | $kJ/h·m^2$ |
| Ground reflectance | 0.2 | - |
| Collector slope | 45 | degrees |

Figure 12 shows the solar heating system simulation process using TRNSYS 16.1 software. In the mentioned model, Type 109 is used to enter the weather conditions into the model. Type 3 is also a pumping system, one used for circulation between the storage source and the solar collector, and the other to transfer the hot water produced to the underfloor heating pipes. Type 5b is a heat exchanger with an asymmetric current. A heat exchanger with an apparent heat capacity of zero is modeled as a fixed-effect device that operates independently of the system configuration. The input temperature of the cold side and the hot side and the flow rate are given as input. It should be noted that the effect is calculated for a constant value of all heat transfer coefficients. Type 6 is a side heater.

The lateral heating element is modeled to increase the fluid flow temperature using an internal, external, or combination of both controllers. Whenever the input of the external controller is equal to one or the output temperature of the heater is less than the value set by the user, the heater adds heat to the flow at the rate designed by the user ($Q_{max}$). Defining a constant value for the control function and specifying a value large enough for $Q_{max}$ will work similar to a home-side water heater with an internal controller to generate the output temperature. With a control function with a value of zero or one of a thermostat or controller, this operation will proceed like a furnace that adds heat at a rate of $Q_{max}$, but the heat increases to the point where the outlet temperature is higher than the test value The storage source, which is Type 4, is also used to store water. It should be noted that at all collector cross sections, when the heating system is not working and we need heating, the solar heating system is producing energy and this productive energy is not consumed and is wasted. These values should be shown in the calculations as zero. Excel software is entered, and the corresponding values are deducted from the total output. These estimates are evaluated annually and hourly at all sections. The very important point in this section is that, when the poultry farm is closed, the distance between the two cycles, which is between 16 to 17 days, produces a solar water heater, which is unusable because, at that time, we do not have poultry house heating if the operation of solar energy has more production than consumption, then it is stored in this tank.

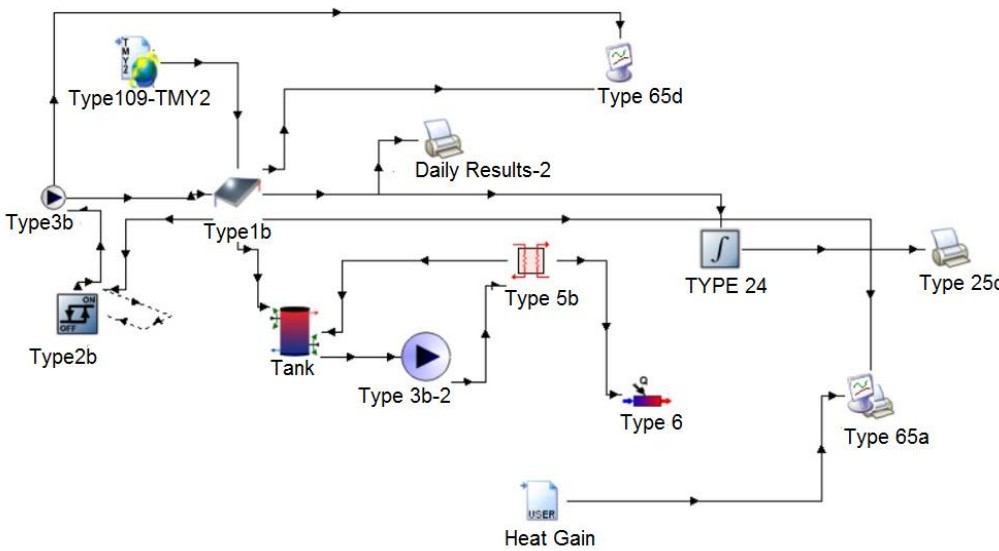

**Figure 12.** Diagram of the simulated process of the solar heating system in TRNSYS software.

It should be noted that in all collector cross sections, when the heating system does not work when we need heating, the solar heating system produces energy and this energy is not consumed, and in the term, is thrown. These values must be shown as zero in the calculations. The output data of this section is entered into Excel 2016 software and the relevant values are deducted from the total output. These estimates are evaluated annually and hourly in all sections. An important point in this section is that when the poultry farm is closed, the interval between two cycles, which is between 16 to 17 days, produces a solar water heater, which is unusable. As, at that time, we do not need to heat the poultry house; if the operation of solar energy has more production than consumption, it will be stored in this tank.

To optimize the cross-section of the solar collector, the Barley and Balcombe method is used [37]. In this way, the SF is calculated first. SF is the amount of energy extracted from the sun in proportion to the total energy required. This expression shows how much percent of the total energy is obtained by solar energy. In this way, throughout the year, the energy extracted from the sun is calculated hour by hour, then the amount of SF is evaluated. These calculations are evaluated for the cross sections and the number of different collectors [37].

Table 6 shows the amount of solar energy gained, the required auxiliary energy, the total amount of energy required, and the SF for the different areas of the solar collector. Accordingly, if we install 2 m$^2$ (a solar collector), we can supply only 5.2% of the total energy by solar energy, and if we install 30 m$^2$ of solar collector, we will be able to receive 100% of the required energy by solar energy.

**Table 6.** Summary of production energy, auxiliary, the total energy required, and SF for different collectors.

| Collector Area (m$^2$) | Solar Energy Gain (kJ/h) | Auxiliary Heating Required (kJ/h) | Total Energy Demand (kJ/h) | SF (%) |
|---|---|---|---|---|
| 2 | 4,941,533 ± 1.2% | 89,730,095 ± 1.2% | 94,671,628 | 5.2% |
| 6 | 15,745,096 ± 1.2% | 78,926,532 ± 1.2% | 94,671,628 | 16.6% |
| 8 | 21,769,296 ± 1.3% | 72,902,332 ± 1.3% | 94,671,628 | 23.0% |
| 10 | 28,209,231 ± 1.5% | 66,462,397 ± 1.3% | 94,671,628 | 29.8% |
| 16 | 50,514,442 ± 1.1% | 44,157,186 ± 1.4% | 94,671,628 | 53.4% |
| 18 | 58,713,939 ± 1.1% | 35,957,689 ± 1.1% | 94,671,628 | 62.0% |
| 20 | 67,086,856 ± 1.1% | 27,584,772 ± 1.1% | 94,671,628 | 70.9% |
| 26 | 93,216,604 ± 1.1% | 1,455,024 ± 1.1% | 94,671,628 | 98.5% |
| 30 | 94,671,628 ± 1.3% | 0 | 94,671,628 | 100.0% |

Figure 13 shows the SF value for different areas of solar collectors. As it can be seen, as the collector area increases, the SF increases. Increasing the collector area leads to an increase in the solar heat flux by the plate, which ultimately increases the SF. The behavior of the diagram up to 26 m$^2$ is quasi-linear, which means that using any collector area up to 26 m$^2$ can significantly increase the SF. As the collector area increases more, the SF increases with less sensitivity; thus, in an area of 26 m$^2$, the SF is equal to 98.5% and in an area of 30 m$^2$, the SF is equal to 100%. By increasing the area, the energy received by the solar collector increases, and the temperature increases. The results show that SF increases with the increase in an area [38,39]. The technically optimal point is the point at which we have maximum production from the sun. That is the point that covers the surface of 26 m$^2$ of the sun. Therefore, by increasing the area by more than 26 m$^2$, only the cost increases.

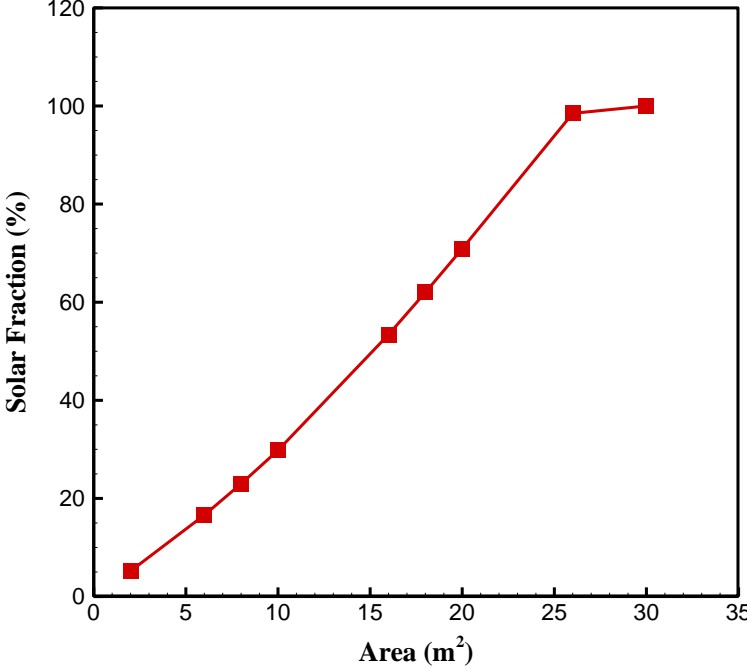

**Figure 13.** SF curve for different collector areas.

Figure 14 shows the curve of the amount extracted from solar energy versus the amount of energy required in the auxiliary system for different areas of solar collectors. As shown in the figure, the intersection of these two curves, which is located approximately 16 m$^2$ of the solar collector, is where half of the energy is supplied by the sun and the other half by the auxiliary system. In the past, this point was considered the optimal point. If this is not the true optimal point, increasing solar energy gain is proportional to increasing the collector area. This curve fits with the SF. As the collector area increases, the solar energy gain increases. The auxiliary heating required behavior is exactly the opposite of the solar energy gain curve so its value decreases as the collector area increases. When the collector area increases, although it increases the cost, it reduces the need to use an auxiliary heating source and it is easy to use a renewable and clean source with a slight increase in cost. From these two curves, it is easy to strike a balance between natural and auxiliary energy sources according to the conditions. Both curves show nonlinear behavior. However, up to 10 m$^2$ of surface, the trend of energy extracted from the sun is uniform, and, from 10 m$^2$ to 16 m$^2$, there is a sudden increase. There is also a linear trend from 16 m$^2$ to 20 m$^2$ and with increasing area, a sudden jump in the amount of energy extracted be seen.

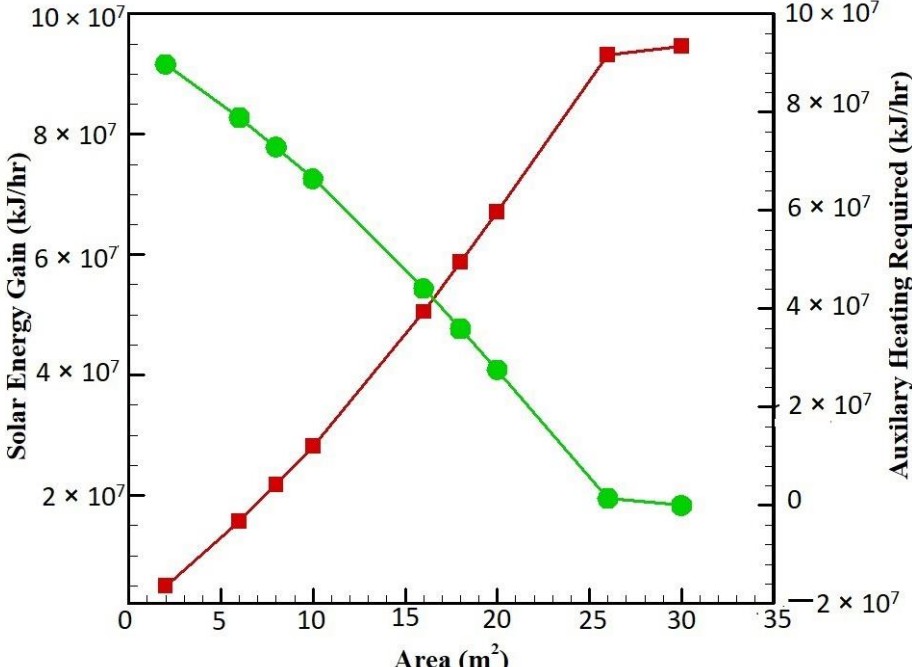

**Figure 14.** The curve of the amount of energy produced by the collector and the amount of auxiliary energy required for different collector areas.

In order to select the optimal area, another point must be considered. This point is known as economic optimization. The point is that investing at this point, in terms of engineering economics, has the highest IRR (internal rate of return equivalent to the rate of return that an investor can earn by investing in a project) or NPV. In this section, this optimization is completed according to IRR, then a comprehensive engineering economics analysis will be performed on the selected optimal area. Figure 15 shows the amount of annual income in Dollars for different areas of collectors in m$^2$. The linear trend is seen up to the area of 26 m$^2$, then the slope of the chart is fixed. Figure 16 shows the investment required for each of the selected areas. This curve is drawn in terms of collector area per square meter, and investment cost in terms of Dollars. The IRR amount of the selected collectors can be obtained according to the planned investment cost and annual income. The linear trend indicates that the increase in the collector area is proportional to the increase in cost. Figure 17 shows the IRR value in percentage for different areas of the collector. As shown in the Figure above, the maximum IRR value has dropped to

26 m$^2$. Therefore, technical and economic optimism has fallen exactly at the same point. As the area increases, the IRR value begins to decrease. The minimum IRR value occurs per minimum area. From 6 m$^2$ to 25 m$^2$, a linear trend for IRR is evident. In an earlier study, the performance of PVT systems, solar collectors, heat pumps, etc., to provide lighting and heating energy for a poultry farm was investigated [20]. The study showed that more than 85% of the energy savings of this poultry farm is provided by this system, and this system can have a return-on-investment equivalent to 3 to 5 years. The main difference between that research and this research is that, first, in the other research [40], the conditions are not seen transiently, also the city under study is different. The difference between the studied cities affects the intensity of ambient radiation to extract thermal energy from the sun. In addition, in this study, no research has been completed on thermal energy storage [40].

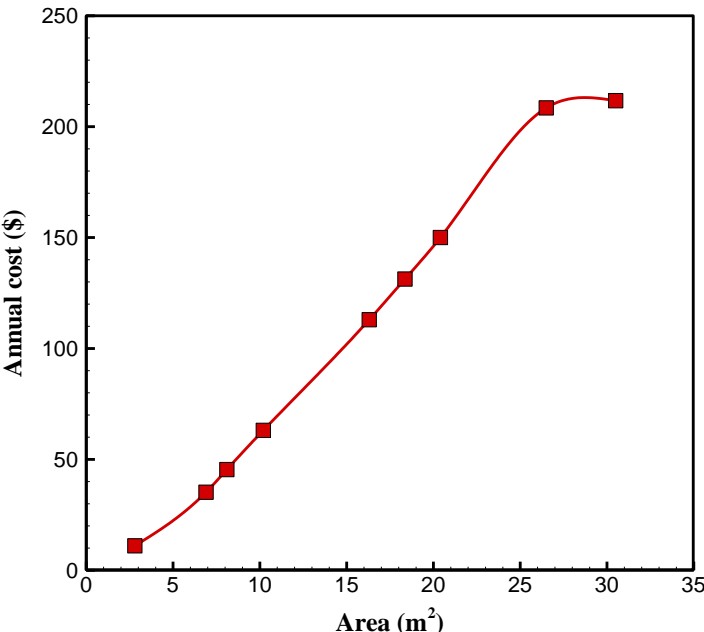

**Figure 15.** Amount of annual cost for a different area of collector areas.

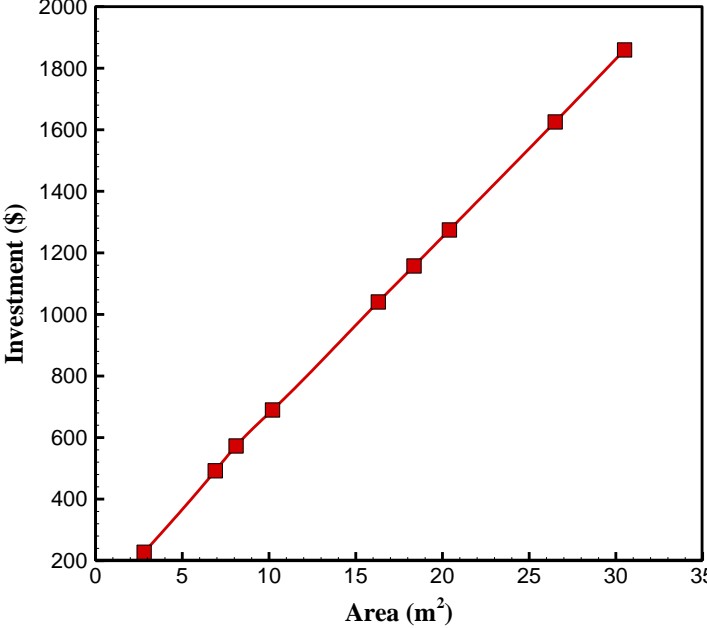

**Figure 16.** Investment curve versus various cross sections of collector areas.

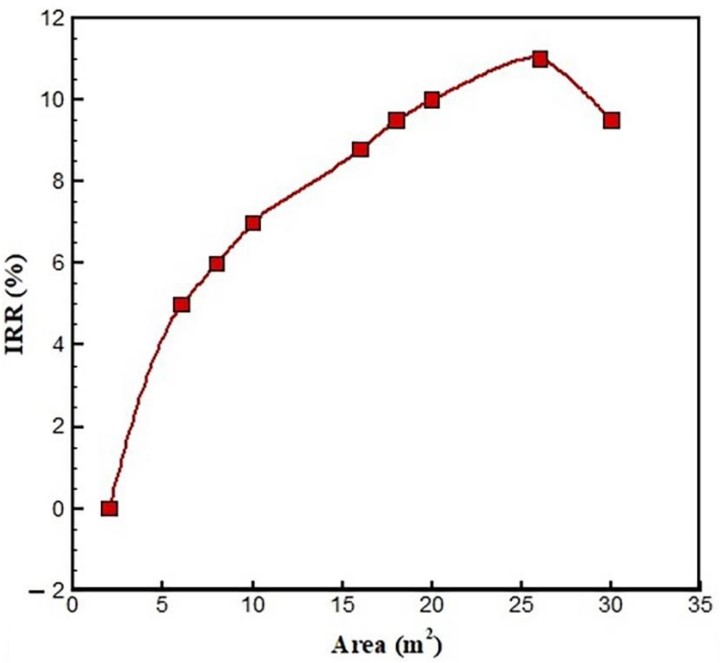

**Figure 17.** IRR value for different collector areas.

Table 7 shows a summary of the calculations presented in this section.

**Table 7.** Summary of the calculations made to optimize the area.

| Collector Area (m$^2$) | Solar Energy Gain (KJ/h) | Auxiliary Heating Required (KJ/h) | Total Energy Demand (KJ/h) | SF (%) | IRR (%) |
|---|---|---|---|---|---|
| 2 | 4,941,533 ± 1.5% | 89,730,095 ± 1.1% | 94,671,628 | 5.2% | 0% |
| 6 | 15,745,096 ± 1.5% | 78,926,532 ± 1% | 94,671,628 | 16.6% | 4.56% |
| 8 | 21,769,296 ± 1.1% | 72,902,332 ± 1% | 94,671,628 | 23.0% | 5.69% |
| 10 | 28,209,231 ± 1.2% | 66,462,397 ± 1.2% | 94,671,628 | 29.8% | 6.60% |
| 16 | 50,514,442 ± 1.3% | 44,157,186 ± 1.3% | 94,671,628 | 53.4% | 8.87% |
| 18 | 58,713,939 ± 1.3% | 35,957,689 ± 1.5% | 94,671,628 | 62.0% | 9.49% |
| 20 | 67,086,856 ± 1.2% | 27,584,772 ± 1% | 94,671,628 | 70.9% | 10.03% |
| 26 | 93,216,604 ± 1.1% | 1,455,024 ± 1% | 94,671,628 | 98.5% | 11.33% |
| 30 | 94,671,628 ± 1.1% | 0 | 94,671,628 | 100.0% | 9.55% |

It is necessary to optimize the slope value. The results of the researchers showed that the slope of the collector affects energy consumption [41,42]. Figure 18 also shows the amount of energy extracted from the sun relative to different slopes. As it can be seen, as the slope increases to 47%, the total solar energy received increases.

It was shown that the inclination angle strongly affects the performance of the collector. The average heat transfer coefficient decreases with increasing inclination angle. This fact suggests that too high an inclination angle is not recommended for solar collectors [43]. A slope of 47% is the best case as it can use solar energy, and the efficiency of the system can be at this maximum. With the increasing slope, more than 47%, the received energy decreases. The use of sloping surfaces can lead to more radiation in the Ardestan region, especially in the cold months of the year.

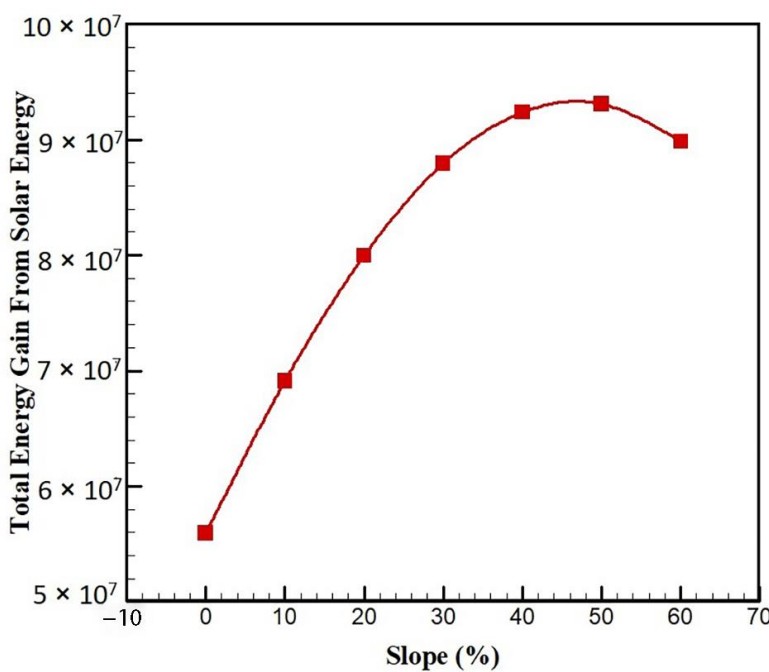

**Figure 18.** Total energy extractable from the sun in different slopes.

In the slope optimization section, first, the amount of total energy that can be extracted from the sun has been measured in different slopes, and their results are presented in Table 8. As shown in Figure 18, the optimal slope in this land-use occurs at 47 degrees in this area. The initial 45 degrees conjecture is very close to reality and the calculations in the previous step are sufficiently valid. In the following, the optimal flow rate through the solar system is calculated. For this purpose, for a surface of 26 m$^2$ and a slope of 47 degrees, we pass different flow rates and estimate the amount of energy extracted from the sun, and, finally, the maximum point is the appropriate flow rate for the pump.

**Table 8.** The total energy that can be extracted from the sun at different slopes. ($\pm 1.1\%$–$\pm 1.6\%$).

| Slope | 0 | 10 | 20 | 30 | 40 | 50 | 60 |
|---|---|---|---|---|---|---|---|
| Solar Useful Gain (kJ/year) | 38,783,668 | 47,211,512 | 53,923,892 | 58,849,719 | 61,601,237 | 62,314,985 | 60,778,776 |
| Total Solar Gain with Reservoir (kJ/year) | 17,160,401 | 21,907,294 | 26,035,076 | 29,103,670 | 30,756,442 | 30,787,288 | 29,075,727 |
| Total Energy Gain From Solar Energy (kJ/year) | 55,944,069 | 69,118,807 | 79,958,968 | 87,953,388 | 92,357,680 | 93,102,273 | 89,854,504 |

Volumetric and mass flow rates have an impact on the performance of solar collectors. Different flow rates cause different efficiencies. Accordingly, choosing the optimum flow rate in every collector should be considered one of the significant factors in selecting and using the collectors. There are many studies about the effect of flow rate on a collector's efficiency [44,45]. A higher flow rate leads to greater efficiency, but a higher flow rate necessitates higher pump power and a larger tube diameter (to maintain laminar flow), both of which raise installation costs. As a result, it is necessary to control and obtain the appropriate flow rate. In the mass flow optimization section of the pump, first, the amount of total energy extractable from the sun in different discharges has been measured, and their results are shown in Figure 19. As the flow rate increases to 1700 kg/h, the total energy received from the sun increases, then a relative decrease is observed. However, the effect

of increasing the flow on the total energy received from the sun to a flow of 1200 kg/hr is very significant and is observed after a gentle upward trend. However, the two main factors of slope and flow rate can be effective in receiving the total solar energy, the role of flow rate is more tangible. As shown in Figure 19, the optimal flow occurs in this use and this area at the point of 1700 kg per hour. According to this case, the optimal flow of the pump is selected, and the pump is placed in the optimal position. Individual components are needed recalculate this collector area with optimal conditions and calculate the volume of the new tank.

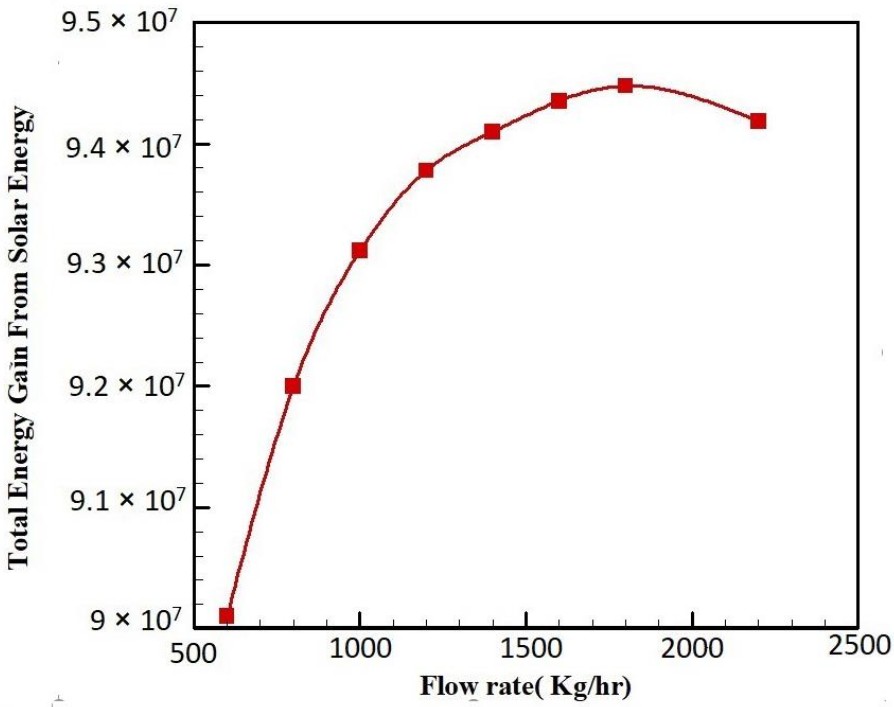

**Figure 19.** Total energy extractable from the sun in different discharges.

In the mass flow optimization section of the pump, first, the amount of total energy that can be extracted from the sun has been measured at different flow rates, and their results are presented in Table 9.

**Table 9.** The total energy that can be extracted from the sun in different flow rates ($\pm 1.1\%$, $\pm 1.7\%$).

| Flowrate | 600 | 800 | 1000 | 1200 | 1400 | 1800 | 2200 |
|---|---|---|---|---|---|---|---|
| Solar Useful Gain (kJ/year) | 60,691,378 | 61,683,121 | 62,243,406 | 62,545,044 | 62,637,207 | 62,650,223 | 62,193,985 |
| Total Solar Gain with Reservoir (kJ/year) | 29,409,522 | 30,312,603 | 30,872,070 | 31,233,406 | 31,461,596 | 31,826,349 | 31,989,438 |
| Total Energy Gain From Solar Energy (kJ/year) | 90,100,899 | 91,995,723 | 1000 | 93,778,449 | 94,098,803 | 94,476,572 | 94,183,423 |

Table 10 shows the overall conclusion for this system. As it is clear in the table, with the final optimization measures, the solar fraction has increased from 98.5% to 99.7%. The volume of the tank is also equal to 440 L. With these interpretations, there is almost no need for auxiliary energy.

**Table 10.** The most optimal suggested mode for the investigated collectors.

| | | |
|---|---|---|
| Collector Area | 26 | m$^2$ |
| Heating Load | 94,671,628 | kJ/year |
| Solar Useful Gain | 62,661,711 | kJ/year |
| Auxiliary Heating Required | 32,009,917 | kJ/year |
| Energy Reservoir in Tank | 31,750,410 | kJ/year |
| Inlet Temperature for Under Floor Heating | 106.4 | F |
| Outlet Temperature for Under Floor Heating | 91.4 | F |
| Temperature Difference | 15 | F |
| Tank Volume | 0.44 | m$^3$ |
| Total Solar Gain with Reservoir | 94,412,121.3 | kJ/year |
| Solar Fraction | 99.7% | % |
| Total NG Consumption | 11,834 | m$^3$/year |
| Total NG Saving by Solar | 11,802 | m$^3$/year |
| Total Auxiliary Heater NG Consumption | 32 | m$^3$/year |

*Simulating the Photovoltaic System to Supply the Hall with Electricity*

In this part of the research, the photovoltaic system is investigated to supply electricity to the hall. It should be noted that, since the price of buying electricity from the grid is much cheaper than the selling price of photovoltaic electricity, in this project, the entire produced electricity is sold to the grid and the required electricity of the complex is purchased from the grid. Figure 20 shows the process of the system under study in TRNSYS software.

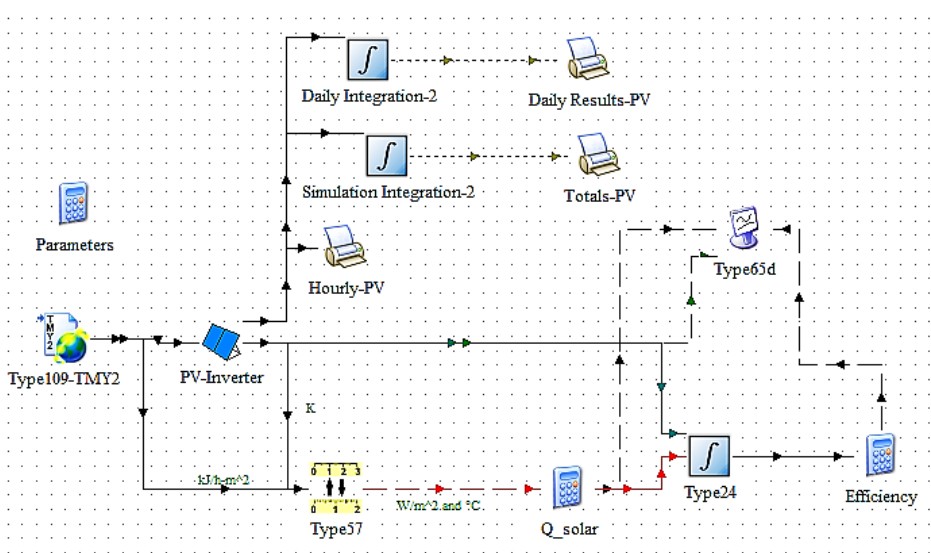

**Figure 20.** Process of photovoltaic system in TRNSYS software.

The technical specifications of the desired photovoltaic are presented in Table 11. It should be noted that this type of monocrystalline silicon system, which is the most widely used and commercial type of photovoltaic cell, has been used.

**Table 11.** The technical specifications of the desired photovoltaic.

| | | |
|---|---|---|
| PV | 106 | W |
| Module short-circuit current at reference conditions | 4.9 | amperes |
| Module open-circuit voltage at reference conditions | 21.6 | V |
| Reference temperature | 298 | K |
| Reference insolation | 1000 | W/m$^2$ |
| Module voltage at max power point and reference conditions | 17 | V |
| Module current at max power point and reference conditions | 5.9 | amperes |

**Table 11.** *Cont.*

| | | |
|---|---|---|
| Temperature coefficient of Isc at (ref. cond.) | 0.02 | any |
| Temperature coefficient of Voc (ref. cond.) | −0.079 | any |
| Number of cells wired in series | 36 | - |
| Number of modules in series | Variable | - |
| Number of modules in parallel | Variable | - |
| Module temperature at NOCT | 313 | K |
| Ambient temperature at NOCT | 293 | K |
| Insolation at NOCT | 800 | W/m$^2$ |
| Module area | 0.89 | m$^2$ |
| Tau-alpha product for normal incidence | 0.95 | - |
| Semiconductor bandgap | 1.12 | any |
| Value of parameter at reference conditions | 1.9 | - |
| Value of parameter I_L at reference conditions | 5.4 | amperes |
| Value of parameter I_0 at reference conditions | 0 | amperes |
| Module series resistance | 0.5 | - |
| Shunt resistance at reference conditions | 16 | - |
| Extinction coefficient-thickness product of cover | 0.008 | - |
| Total incident radiation on tilted surface | Variable | kJ/h·m$^2$ |
| Ambient temperature | Variable | C |
| Load voltage | Variable | V |
| Array slope | Variable | degrees |
| Beam radiation on tilted surface | Variable | kJ/h·m$^2$ |
| Sky diffuse radiation on tilted surface | Variable | kJ/h·m$^2$ |
| Ground diffuse radiation on tilted surface | Variable | kJ/h·m$^2$ |
| Incidence angle on tilted surface | Variable | degrees |
| Solar zenith angle | Variable | degrees |
| Wind speed | Variable | m/s |

Now, the amount of electricity required in different parts of the hall is examined. These sections are lighting, underfloor heating pump, ventilation fan, auxiliary burner fan, solar water heater pump, and miscellaneous applications, the process of which is shown in Figure 21.

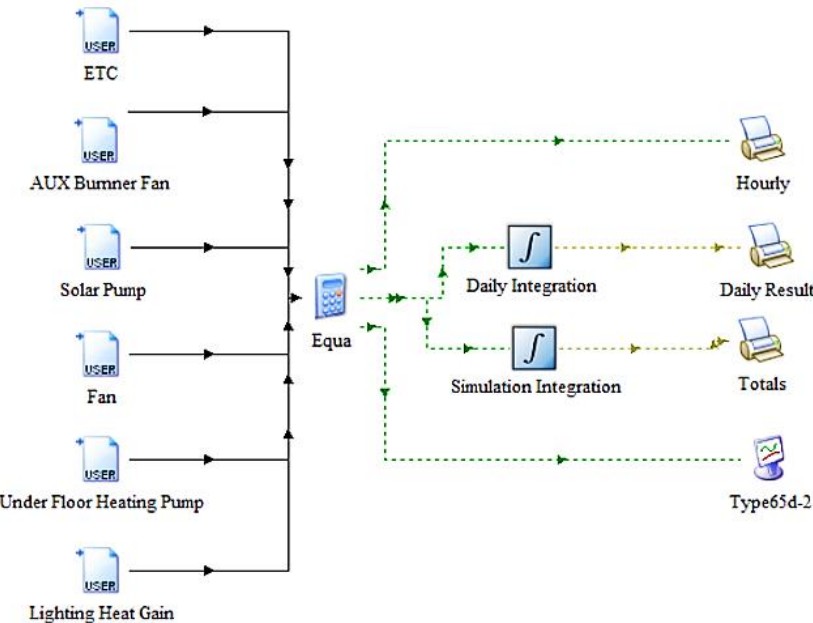

**Figure 21.** Checking the amount of electricity required in different parts of the hall in TRNSYS software.

Table 12 shows the technical specifications of the mechanical equipment used in the system.

**Table 12.** The technical specifications of the mechanical equipment used in the system.

| | | | |
|---|---|---|---|
| Collector specifications | Number of solar collectors | 13 | - |
| | Area of each unit | 3 | m$^2$ |
| | Total area | 26 | m$^2$ |
| Specifications of the tank | Volume | 0.44 | m$^3$ |
| Specifications of pump | The volumetric flow rate of the solar pump | 1700 | Kg/h |
| | The volume flow rate of the solar cycle pump | 1.7 | m$^3$/h |
| | Solar pump head | 8 | m |
| | Power consumption | 245 | W |
| Specifications of floor heating pump | Floor heating pump | 1500 | W |
| Burner auxiliary system | Burner power consumption | 250 | W |
| Fan specifications | Flow rate | 600 | m$^3$/h |
| | Head | 0.5 | in H$_2$O |
| | Power consumption | 370 | W |
| | Number of fans | 2 | - |
| | Total power consumption | 740 | W |
| The total power consumption | | 2735 | W |

The area of each photovoltaic module is equal to 0.89 m$^2$. Therefore, in this section, the number of 30 photovoltaic modules is examined. It should be noted that 10 cells have been used in series in 3 parallel circuits. In addition, according to the manufacturer's recommendation, 10 series are allowed in each parallel circuit in all cross-sections of the mentioned photovoltaic systems [46]. It shows the summary of the results of using the photovoltaic system. Table 13 shows the amount of electricity produced by photovoltaics, the amount of electricity required in the hall, the purchase price of electricity, the sale price of electricity, the annual income, the required investment cost, and, finally, the IRR index for the cross-sectional area.

**Table 13.** Summary of the results of using the photovoltaic system ($\pm$1.2%–$\pm$1.6%).

| Total Area | Total Power Production (kWh·Year) | Total Power Need (kWh·Year) | Power Purchase Price ($) | Power Network Price ($) | Annual Revenue ($) | Investment ($) | IRR (%) |
|---|---|---|---|---|---|---|---|
| 26.7 | 6589 | 14,566 | 0.0416 | 0.004 | 215.8 | 3204 | 3.0% |
| 44.5 | 10,981 | 14,566 | 0.0416 | 0.004 | 398.5 | 5340 | 4.2% |
| 62.3 | 15,373 | 14,566 | 0.0416 | 0.004 | 581.3 | 7476 | 4.6% |
| 89 | 21,962 | 14,566 | 0.0416 | 0.004 | 855.4 | 10,680 | 5.0% |
| 178 | 43,924 | 14,566 | 0.0416 | 0.004 | 1769.0 | 21,360 | 5.4% |
| 222.5 | 54,905 | 14,566 | 0.0416 | 0.004 | 2225.8 | 26,700 | 5.5% |

Based on this, the graph of Figure 22 is presented based on the amount of total power production for a different area of the photovoltaic system.

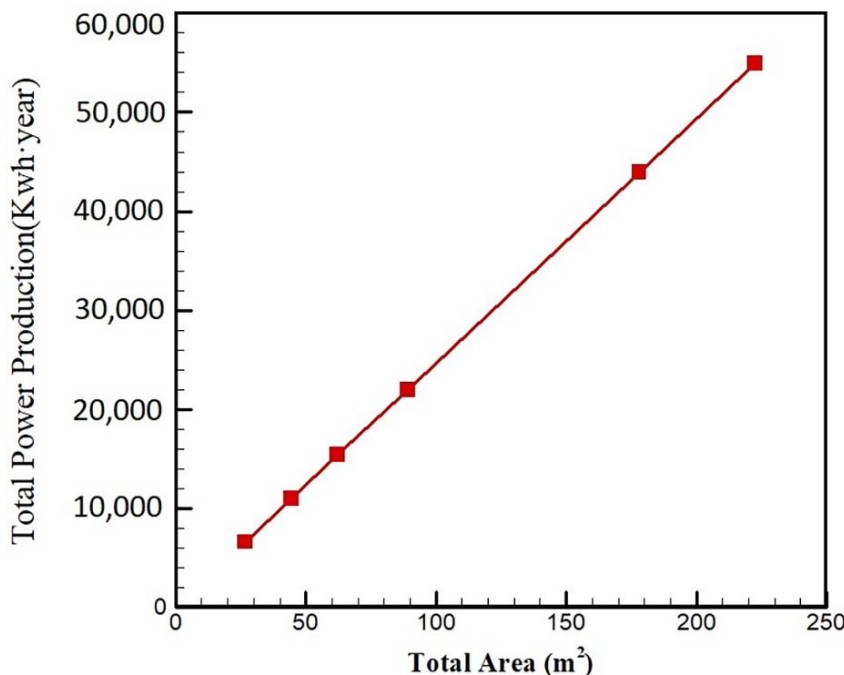

**Figure 22.** Total Power Production in terms of total area.

Figure 23 is presented based on the investment cost of the system for a different area of the photovoltaic system.

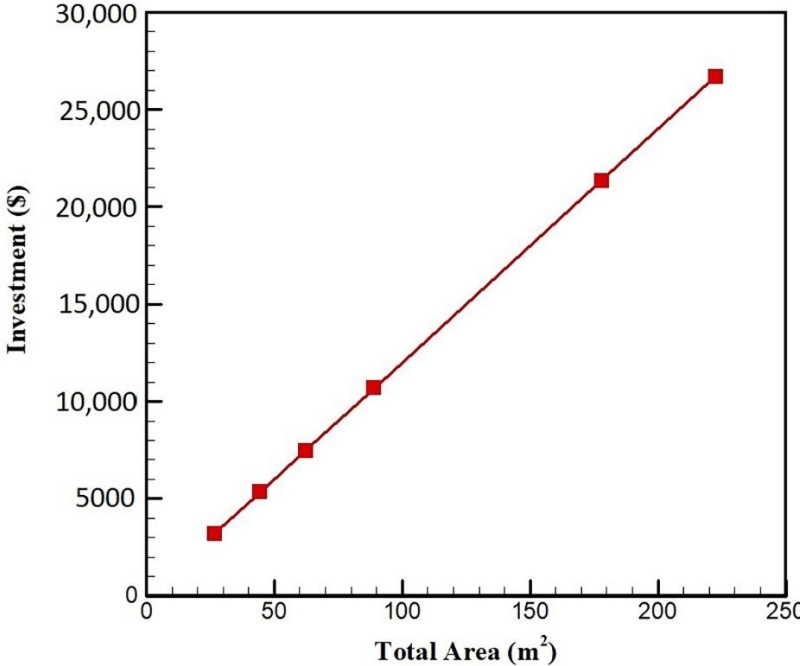

**Figure 23.** The investment cost of the system for a different area of the photovoltaic system.

Figure 24 shows the annual income for a different area of the photovoltaic.

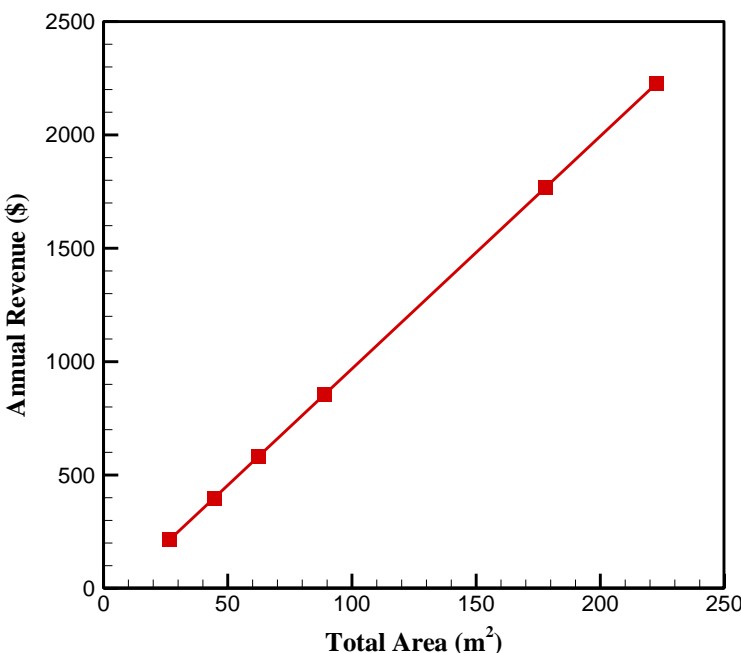

**Figure 24.** Annual income for a different area of the photovoltaic system.

Figure 25 shows the IRR rate for a different area of the photovoltaic system.

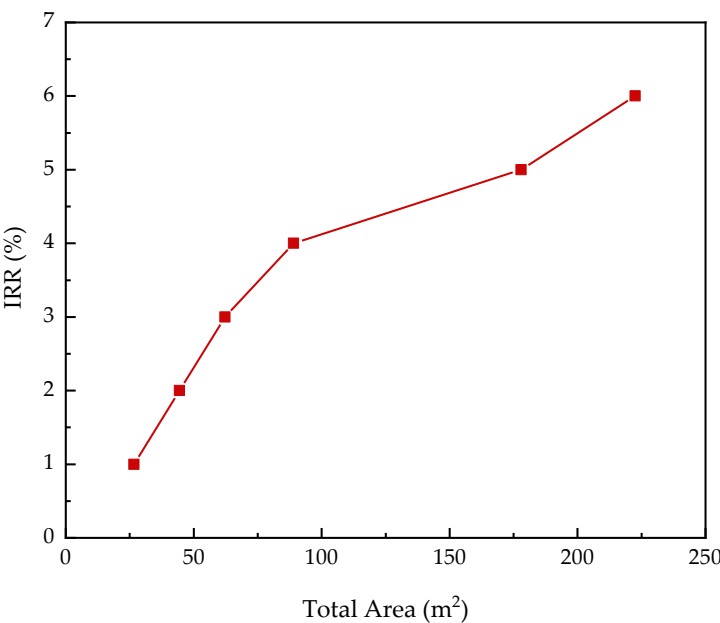

**Figure 25.** The IRR for a different area of the photovoltaic system.

As it is clear in the IRR curve, the optimal point is in the section where the IRR is most upward-sloping. This point shows the number of 6 circuits with 10 series modules, a total of 60 with an area of 53.4 m$^2$. In the following section, the calculations of the optimal cross-sectional area for photovoltaic cells are discussed. The review is presented in Table 13. Table 14 summarizes the optimal conditions for the photovoltaic system under study.

**Table 14.** Optimal conditions for photovoltaic system.

| | | |
|---|---|---|
| Number of Panel in series | 10 | - |
| Number of Panels in Parallel | 6 | - |
| Each Panel Area | 0.89 | $m^2$ |
| Total Number of Panel | 60 | - |
| Total Panel Area | 53.4 | $m^2$ |
| Annual Power Production | $13{,}177 \pm 1.1\%$ | kWh·year |
| Total Power Need | $14{,}566 \pm 1.1\%$ | kWh·year |
| Power Purchase Price | 0.0416 | $ |
| Power Network Price | 0.004 | $ |
| Annual Revenue | 489.9 | $ |
| Investment | $6408 \pm 1.1\%$ | $ |
| IRR | $5 \pm 0.1\%$ | % |

To validate the calculations made in this section, using PVSOL software, the mentioned photovoltaic system was designed, and the results of the calculations are presented below. For this purpose, 60 panels with similar specifications to the simulated model in TRNSYS software were evaluated. The diagram in Figure 26 of the proposed PVSOL software model is based on this.

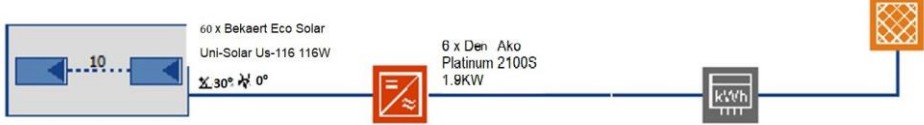

**Figure 26.** Diagram of the proposed PVSOL model.

As shown in Figure 26, 10 panels are used in parallel and a total of 60 panels (6 parallel circuits) are used. Based on this, the annual energy production chart was evaluated, and it was found that the amount of energy produced by these panels is equal to 13,116 kWh per year. This value is estimated at 14,247 kWh in TRNSYS software. The deviation of these two software from each other is less than 1%, so the accuracy of calculations is high and acceptable. Figure 27 shows the power generation curve in PV and inverter systems. As can be seen, the energy from the inverter is always more than the energy produced from the PV array in all months. For both cases, the maximum and minimum system variant is related to the 8th and 11th months, respectively. In the first quarter of the year and the third quarter of the year, the changes are increasing in a regular pattern. However, in the fourth and tenth months, the variant system decreases and in the eleventh month, a sharp decline is seen. In both cases, the third quarter of the year seems to have a high potential and the fourth quarter of the year has the lowest potential.

Table 15 shows the results of these calculations for the areas of 2 $m^2$, 4 $m^2$, and 6 $m^2$, respectively.

Due to the electricity purchase rate being much lower than the renewable electricity sale rate to the grid, the scenario of selling the entire electricity to the grid and buying electricity separately was proposed. Based on this, the number of 60 solar panels in 6 parallel sections, each of which had 10 in series, was identified as the best case. Economic analysis of this solution using RET Screen Expert Professional 8.0.1.31, this solution is not economical under any heading due to the IRR below the bank rate and negative NPV.

Among the research that have been completed, very few studies have been completed in relation to energy storage in poultry farms.

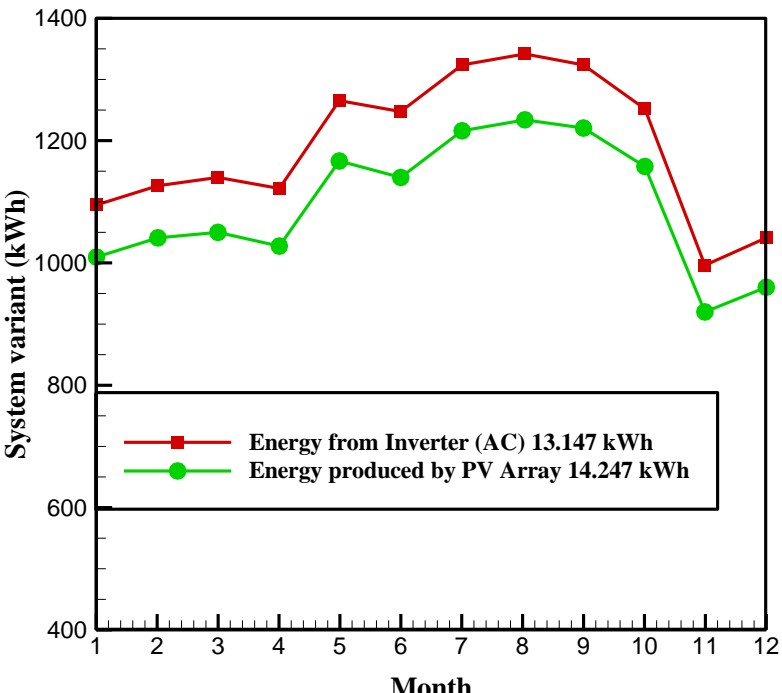

**Figure 27.** Electric power generation curve in PV and system inverter.

**Table 15.** Results from the use of the surface of 2, 6, and 8 m$^2$.

| Property | Value | Value | Value | Unit |
|---|---|---|---|---|
| Collector Area | 2 | 6 | 8 | m$^2$ |
| Heating Load | 94,671,628 | 94,671,628 | 94,671,628 | kJ·year$^{-1}$ |
| Solar Useful Gain | 4,786,686 | 14,360,407 | 19,147,267 | kJ·year$^{-1}$ |
| Auxiliary Heating Required | 89,884,942 | 80,311,221 | 75,524,361 | kJ·year$^{-1}$ |
| Energy Reservoir in Tank | 154,847 | 1,384,689 | 2,622,029 | kJ·year$^{-1}$ |
| Inlet Temperature for Under Floor Heating | 106.4 | 106.4 | 106.4 | F |
| Outlet Temperature for Under Floor Heating | 91.4 | 91.4 | 91.4 | F |
| Temperature Difference | 15 | 15 | 15 | F |
| Tank Volume | 0.03 | 0.1 | 0.13 | m$^3$ |
| Total Solar Gain with Reservoir | 4,941,533.1 | 15,745,095.5 | 21,769,295.9 | kJ·year$^{-1}$ |
| SF | 5.2 | 16.6 | 23 | % |
| Total NG Consumption | 11,834 | 11,834 | 11834 | m$^3$·year$^{-1}$ |
| Total NG Saving by Solar | 618 | 1968 | 2721 | m$^3$·year$^{-1}$ |
| Total Auxiliary Heater NG Consumption | 11,216 | 9866 | 9113 | m$^3$·year$^{-1}$ |
| NG Price | 0.18 | 0.18 | 0.18 | $ |
| Annual Cost Without Solar Heater | 211.72 | 211.72 | 211.72 | $·year$^{-1}$ |
| Annual Auxiliary Heater Cost | 200.66 | 176.50 | 163.03 | $·year$^{-1}$ |
| Annual Solar Heater Saving | 11.05 | 35.21 | 48.68 | $·year$^{-1}$ |

Dbuok et al. [34] investigated the solar-heated poultry house in BEKAA town. The results showed that the tested system managed to provide the different temperatures required for rearing chickens during the winter of 2019, where the temperature ranged from 32 °C during week 1 to 18 °C during week 6. This solar heating system with a cross-sectional area of 6 m$^2$ saves 60% of heating costs when using traditional heating methods. In the present research, the cross-sectional area of 18 m$^2$ in the city of Ardestan with the mentioned poultry farming conditions has a saving value of 60%. The reasons for the difference between these two poultry farms are as follows:

- The first difference investigated in the village of Beka in Lebanon is the climatic conditions of the region. According to the investigation, the Heating Degree Days in Bekah city is equal to 443 days, while in Ardestan city, it is equal to 2038 days.

- The difference in weather conditions is another reason for the difference caused by these two chicken farms.
- The difference in the dimensions of the hall and the number of poultry used in the poultry farm.
- Another difference between this system and the system investigated in this research is the use of radiators instead of floor heating. The underfloor heating system can work at a lower temperature. Also, because this system spreads evenly throughout the building, it has much higher comfort and efficiency compared to the radiator system.

Li et al. [47] investigated the performance of PVT systems, solar collectors, heat pumps, etc. to provide lighting and heating energy for a chicken farm. Their study showed that more than 85% of the savings in the energy consumption of this poultry farm was provided by this system and this system can have a return-on-investment equivalent to 3 to 5 years. The main difference between the previous research and this research is that the conditions were not seen transiently, and the city under investigation is different. The difference between the investigated cities affects the intensity of ambient radiation to extract thermal energy from the sun. In addition, in this research, there has been no investigation regarding thermal energy storage.

Also, in other studies that have been completed, the role of the angle and surface of the collector in solar fraction has been investigated. Table 16 shows the comparison between the research results and this research.

**Table 16.** The comparison between the research results and this research.

| Factor | Housing Type | Main Energyconsuming Activity | Analyzed Energy Type | Area (m²) | Angle (Degree) | SF (%) | Energy Saved (100) | Ref. |
|---|---|---|---|---|---|---|---|---|
| continental climate | Poultry House | Heating and lighting | Natural gas | 8 | NA | NA | 60% | [34] |
| continental climate | poultry house | Heating and lighting | Solar PV systems | 9.22 | NA | NA | 66.79 | [47] |
| continental climate | NA | Heating | Solar collector | 1.1 | NA | 99 | 58% | [38] |
| waste thermal energy | a steam power plant | - | solar combined cooling, heating, and power (CCHP) system | 3000 | NA | 3.2 | 39.9% | [39] |
| solar thermal systems | hybrid power plant | - | Combination of a grid connected photovoltaic (PV) plant with a compressed air energy storage system (CAES) | 1 | NA | - | 17 | [40] |
| continental climate | Poultry House | Heating and lighting | Natural gas | 26 | 47 | 99 | 100 | This work |

NA: Not Available.

According to Table 16, it can be seen that good studies have been completed on the use and application of solar energy for the poultry industry, but most of the studies focus on reducing environmental effects, identifying strategies to improve performance and implementing systems with the aim of supplying electricity to poultry farming, and helping to increase network electricity. In this research, a solar heating system has been designed to reduce energy consumption in a poultry farm (with a capacity of 300 pieces). By using this system, the amount of energy consumption is greatly reduced. The solar water heater in this system is optimized for heating the hall using the floor heating system. In this study, different solar thermal collectors are investigated, and the best solar collector surface is suggested to achieve maximum efficiency of solar energy.

## 4. Conclusions

In this research and modeling, several innovations have not been investigated with this precision in any research so far. These investigations include:

A.   Hour-by-hour dynamic simulation of the chicken breeding hall.
B.   Calculation of the heat produced by the body of chickens with the change of weight and its effect on the heat load of the building with the growth process of the chicken.
C.   Calculating the humidity produced by the body of chickens by changing the weight and its effect on the humidity produced and ventilation of the building with the growing process of chickens.

Considering the mentioned cases in this research, the findings were presented with much higher accuracy.

The results were generally analyzed in two architectural sections, solar thermal system and solar electric system, transiently (hour by hour) in TRNSYS software, then in RETSCREEN software, engineering economics analysis was completed for each. The findings that can be obtained are as follows:

-   A total of $1.37 \times 10^8$ kJ/h of power is needed to heat the study hall for one year (this amount is equal to 17,180 m$^3$ of natural gas).
-   The use of double-glazed windows and insulation for the exterior walls of the building leads to a significant reduction in energy.
-   Using the proposed solutions, the required annual gas consumption can be reduced to 11,833 m$^3$.
-   To achieve optimal conditions, 26 m$^2$ of a solar collector with a slope of 47 degrees (taking into account the volume of the tank 440 L and pump flow of 1700 kg/h) is required.
-   The solar system with 26 m$^2$ and an optimal slope of 47°, and, taking into account the tank volume of 440 lit and the pump flow rate of 1700 kg/h, has the most optimal conditions for providing 100% energy. By analyzing the engineering economy in RETSCREEN software, it was found that the internal interest rate of using this solution, if the hall is equipped with underfloor heating, is equal to 34.1%. If we consider the cost of equipping this hall to install the underfloor heating system, the internal interest rate of this solution is equal to 27.9%.

Optimally for a photovoltaic system, 60 solar panels in 6 parallel sections, each with 10 in series, are required for the network.

In the following, it is suggested to evaluate different types of solar collectors, such as parabolic collectors and EVACUTED TUBE collectors, and the effect of using different collectors in future research. In addition, it is suggested that because in the summer the indoor temperature reached a higher value than the ideal conditions for raising chickens (in this research, 38 °C), the use of cheap cooling systems to increase breeding efficiency should be evaluated and analyzed. In addition, in this research, the effect of preparing the optimal temperature on the efficiency of the herd is not taken into account, which can help to make the designs more economical. This case can also be considered by researchers in future research.

The limitations in this research are mostly related to the lack of scientific and correct exploitation of poultry halls. As this work is completed traditionally, most of the property owners do not have large enough operation to exploit large poultry halls. For example, in the hall, chickens of different ages are not paid special attention, and in addition, due to the use of non-standard heating systems, the farmers cannot have proper control of different environments. These devices, such as jet heaters or rockets and similar things, create heat at one point, then direct it to different areas of the hall utilizing a fan. This causes non-uniform production of heat inside the hall.

In addition to the problem mentioned in the previous section, the problem with halls that are managed traditionally is improper ventilation. Sometimes in these halls, due to

improper ventilation, dust and humidity rise in the hall and deviate from the standard conditions.

Overcrowding, stress, aggressiveness, and high losses are the consequences of these issues. The above cases reduce the efficiency of the herd and mortality, and reduce the profitability of the breeder.

**Author Contributions:** Conceptualization, A.B.; methodology, A.B. and M.J.; software, M.J. and M.M.; validation, A.B., M.J. and B.F.; formal analysis, A.B. and M.J.; investigation, M.J. and B.F.; resources, M.J.; data curation, M.J. and A.B.; writing—original draft preparation, M.J.; writing—review and editing, M.J., A.B. and M.M.; supervision, A.B.; project administration, A.B. All authors have read and agreed to the published version of the manuscript.

**Funding:** This research received no external funding.

**Data Availability Statement:** No data were used for the research described in the article.

**Conflicts of Interest:** The authors declare no conflict of interest.

## Nomenclature

| | |
|---|---|
| $m$ | indoor air mass [kg] |
| $A_f$ | wall and door surface [m$^2$] |
| $C_i$ | specific heat of the air [J/kgK] |
| $C_P$ | specific air heat capacity [J/kgK] |
| $h_{fi}$ | heat transfer coefficient of the fluid $\left[\text{W/m}^2\text{K}\right]$ |
| $Q_{chicken}$ | heat produced by the chicken [W/chicken] |
| $Q_f$ | heat loss from the floor of the building [W] |
| $Q_s$ | sensible heat generated by broiler |
| $Q_{sens}$ | tangible heat produced by the chicken |
| $q_{sup}$ | complementary heat capacity |
| $Q_v$ | heat loss by the ventilation system [J/kgK] |
| $Q_w$ | heat loss from walls and ceiling [W] |
| $R_{is}$ | thermal resistance unit of inner layer surface area $\left[\text{m}^2\text{K/W}\right]$ |
| $R_{os}$ | thermal resistance unit of outer layer surface area $\left[\text{m}^2\text{K/W}\right]$ |
| $R_t$ | total thermal resistance $\left[\text{m}^2\text{K/W}\right]$ |
| $T_i$ | indoor temperature [K] |
| $T_i$ | internal temperature [K] |
| $T_{in}$ | inlet air temperature [K] |
| $T_o$ | outdoor temperature [K] |
| $T_{out}$ | outlet air temperature [K] |
| $\rho_i$ | density of air $\left[\text{kg/m}^3\right]$ |
| $A_c$ | heat transfer path |
| $K_{1,2}$ | thermal conductivity of each layer [W/mK] |
| $R_{1,2}$ | thermal resistance of unit surface area of each building layer $\left[\text{m}^2\text{K/W}\right]$ |
| $X_{1,2}$ | thickness of each layer of the building [m] |
| $m_b$ | Chicken mass [kg] |
| $q_v$ | total volumetric rate of airflow $\left[\text{m}^2\text{/s}\right]$ |
| BLC | Building Load Coefficient |
| F | experimentally constant factor of ambient heat loss [W/mK] |
| P | building environment [m] |
| Qc | heat dissipation through wall or ceiling |
| t | time [s] |
| U | overall heat transfer coefficient $\left[\text{W/m}^2\text{K}\right]$ |

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
