# Peer review of "Reducing Energy Consumption in a Poultry Farm by Designing and Optimizing the Solar Heating/Photovoltaic System"

_sustainability, doi:10.3390/su15076059_

Round 1

Reviewer 1 Report

This research consisted of two main parts; the thermal modeling section and the electrical building modeling section. In the thermal modeling section, the amount of heat required for the building was first modeled using TRNSYS software.

In this study, which is a simulation study, the presence of live animals in the poultry house was not taken into account. Economic and energy consumption should be re-determined, and the comparison should be made by simulating the live animals in the poultry house.

Author Response

Respected Reviewer. Please check the attachment file. Regards

Reviewer 2 Report

Recommendation: Publish after major revisions.

Comments:

This study needs major revision and the following items must be followed;

- All abbreviations must be given in full where first in use.

- Add more results in the abstract.

- What is the main aim of this study? It should be specified at the end of the introduction.

- Introduction should be improved with different Journal papers.

Add articles from last five years

- Since the innovation statement is incomplete, it should be stated which results of this study are original.

- The uncertainty analysis must be performed and tabulated instead.

- The conclusion and discussion section should be supported by previously published articles on the subject.

- The interpretation of the results is weak and a detailed explanation should be given.

- What is the main result of this study? It should be indicated at the end of the conclusion section and a few recommendations must be given for further studies.

- The presentation of known and predicted results must stand well.

- What exactly is the difference between this study and other studies in the literature? The answer to this question should be reflected in the study.

Author Response

(The authors gave the same response as above.)

Reviewer 3 Report

This manuscript presented the reduction of energy consumption in a poultry farm by designing and optimizing the solar heating /photovoltaic system. There are major issues associated with the manuscript which must be addressed before further processing of the paper. 

1. Abstract doesn't provide information about selected problem in clear way. It can be re-written in more comprehensive way. Also, It is suggested to insert some numerical data extracted from your research in the abstract section.

2. Related work should be mentioned in a separate section by highlighting the comparative analysis in tabular manner. What are the unique features of this study compared to the existing works?

3. Contributions should be highlighted in bullet points and justified

4. A ‘Research Gap’ section should incorporate which will states the purpose of the study. 

5. Methodology section is not clear in present form. A flowchart should incorporate which represent the various steps of the proposed work.

6. Incorporate the organization section in separate sub section of the introduction section. 

7. Enhance the quality of the figure 3.

8. Provide the detail description of the design parameters utilized for the development of simulation model presented in figure 3, figure 5, figure 10, figure 11 and figure 12.

9. Authors must clearly present that how much percent reduction obtained in energy consumption. 

10. A comparative analysis with the similar studies and other optimization techniques  are missing which must be incorporated and compared in terms of efficiency enhancement using the proposed work. it is must to show the novelty of the work. 

11. Authors must incorporate the load profile of the selected case study area. 

12. Mathematical modelling of the PV system design must be incorporate in more detailed way such as number of PV modules, inverter, etc. 

13. Manuscript must be reorganized to be present in research paper form. 

14. A separate sub section related to the problem formulation and constraints must be incorporated in the revised version of the manuscript. 

15. More recent references are required to support the novelty of the work. 

16. Authors must submit the manuscript under case study category or review category as novelty is low. 

Author Response

(The authors gave the same response as above.)

Reviewer 4 Report

Review Report for MDPI Sustainability

Manuscript # : sustainability-2193290

Title        : Reducing energy consumption in a poultry farm by designing

                  and optimizing the solar heating /photovoltaic system

Title:

The title looks OK, but you still can optimize it up to 20 words.

Abstract:     

Actually, the abstract is almost perfect, which already has a brief explanation, such as; background, purpose, method, but unfortunately it doesn't show briefly what results have been achieved. I checked, the abstract currently has 199 words, which can still be optimized to 200-250 words, even up to 300 words. To make it more interesting for the reader, I suggest briefly explaining the key results that have been obtained.

Keywords:

Some keywords are still too general (such as: modeling and optimization) and have been used in the title too. To further optimize search engines, both of these should be avoided. I suggest adding, such as; TRNSYS simulation; solar collector, energy modeling

Introduction

The Introduction section needs to be improved. Please consider the following notes.

  • The Introduction section consists of 1 long paragraph. I suggest separating into 3 paragraphs, such as; paragraph for background/overview, paragraph for literature review, and last paragraph for review summary, novelty, gap, purpose, etc.
  • I think, the novelty of this manuscript is still not clear to the readers. For this reason, I suggest concluding your literature review section before stating the purpose of the manuscript.
  • Because you are using the TRNSYS simulation, I suggest adding some references that are very related to the use of TRNSYS in designing heating systems for poultry houses. Please make sure the references are up to date, or not more than 5 years old.
  • For the last paragraph of the Introduction, please cut this part: "In the second part, .............. of the present work is stated." because it's unnecessary.

Materials and Methods:

  • To make it clearer for the reader, please write directly the nomenclature used to explain the formula.
  • For consistency in abbreviating nomenclature and formulas, I suggest abbreviating Solar Fraction to SF.
  • In order to make the manuscript more concise and efficient, I think Figures 2a and 2b can be made side by side.

Results and Discussions:

  • To provide a clearer picture, please complete the detailed specs in tabular form for each Solar Collector and PV panel component, such as; type, construction, efficiency, etc.) used in the TRNSYS simulation and these items should be included in Discussions session. Please adjust the others accordingly.
  • If you can add the thermal efficiency of the solar collector system and the electrical efficiency of the photovoltaic system to the existing graph, that would be even better.
  • The last paragraph of Results and Discussions section seem to mention a bit about the Discussion. However, there is no special paragraph that really explains the Discussion. I suggest to prepare some points for discussion in each paragraph without any more graphs or tables in the discussion. You should discuss some important things related to what you have presented in the results section. I suggest to discuss more in: 1). System design and configuration, 2). System Operation, 3). Investment analysis.
  • Please continue with citations from previous related work into each of the discussion paragraphs.
  • Then, in the last paragraph of the Discussions section, please briefly explain, such as; the practical application and other contributions of this research, then finish with suggestions for further research directions. Relevant and related references should be cited as necessary.

Conclusions:     

The conclusion is OK. Good job. The Conclusions section has briefly explained the results obtained (key findings) and also (should be included) the discussion part of the manuscript.

References:

There are some references that are more than 5 years old, some are even older and 10 years old. I recommend replacing them with the latest ones. For that, I recommend replacing them with a newer one and trying adding other references as needed. I saw in the references there are also some books. I suggest to replace those reference books with peer reviewed journal articles.  

English: Dear Authors. I suggest checking and optimising the English with Grammarly (the free basics version is enough). Especially for this free version, I suggest trying to achieve a minimum score of around 90. (If you check again with the premium version, it will be around 85).

Author Response

(The authors gave the same response as above.)

Round 2

Reviewer 2 Report

The authors have done enough to answer the queries. The article can be accepted. Good Luck.

Author Response

Thanks again. We modified few parts of the manuscript regarding to the reviewers comments that is highlighted.

Reviewer 3 Report

Authors tried to incorporate comments of the reviewer but still some issues are remaining, which must be addressed before further processing of the paper. 

1. Comment number 10 still not properly incorporated. 

2. Comment 11 still not incorporated. 

3. Comment 9 still not clear. 

Reviewer 4 Report

Dear Authors

Thank you for your revision.

Regards,

Reviewer

Author Response

Thanks again. We made a bit changes regarding to the reviewers comments.